# Bud23 promotes the final disassembly of the small subunit Processome in *Saccharomyces cerevisiae*

**Joshua J. Black**[ID]**, Richa Sardana**[ID]**¤, Ezzeddine W. Elmir, Arlen W. Johnson**[ID]*

Department of Molecular Biosciences, The University of Texas at Austin, Austin, Texas, United States of America

¤ Current Address: Department of Molecular Biology and Genetics, Weill Institute for Cell and Molecular Biology, Cornell University, Ithaca, New York, United States of America

* arlen@austin.utexas.edu

**Data Availability Statement:** All relevant data are within the manuscript and its Supporting information files.

## Abstract

The first metastable assembly intermediate of the eukaryotic ribosomal small subunit (SSU) is the SSU Processome, a large complex of RNA and protein factors that is thought to represent an early checkpoint in the assembly pathway. Transition of the SSU Processome towards continued maturation requires the removal of the U3 snoRNA and biogenesis factors as well as ribosomal RNA processing. While the factors that drive these events are largely known, how they do so is not. The methyltransferase Bud23 has a role during this transition, but its function, beyond the nonessential methylation of ribosomal RNA, is not characterized. Here, we have carried out a comprehensive genetic screen to understand Bud23 function. We identified 67 unique extragenic *bud23Δ*-suppressing mutations that mapped to genes encoding the SSU Processome factors *DHR1*, *IMP4*, *UTP2* (*NOP14*), *BMS1* and the SSU protein *RPS28A*. These factors form a physical interaction network that links the binding site of Bud23 to the U3 snoRNA and many of the amino acid substitutions weaken protein-protein and protein-RNA interactions. Importantly, this network links Bud23 to the essential GTPase Bms1, which acts late in the disassembly pathway, and the RNA helicase Dhr1, which catalyzes U3 snoRNA removal. Moreover, particles isolated from cells lacking Bud23 accumulated late SSU Processome factors and ribosomal RNA processing defects. We propose a model in which Bud23 dissociates factors surrounding its binding site to promote SSU Processome progression.

## Author summary

Ribosomes are the molecular machines that synthesize proteins and are composed of a large and a small subunit which carry out the essential functions of polypeptide synthesis and mRNA decoding, respectively. Ribosome production is tightly linked to cellular growth as cells must produce enough ribosomes to meet their protein needs. However, ribosome assembly is a metabolically expensive pathway that must be balanced with other cellular energy needs and regulated accordingly. In eukaryotes, the small subunit

**Funding:** This work was supported by National Institutes of Health (https://www.nih.gov/) grants GM127127, GM053655, and GM108823 to AWJ, a fellowship from the University of Texas at Austin (https://www.utexas.edu/) Graduate School to JJB, and a fellowship from the University of Texas at Austin Office of Undergraduate Research to EWE. The funders had no role in study design, data collection and analysis, decision to publish, or preparation of the manuscript.

**Competing interests:** The authors have declared that no competing interests exist.

(SSU) Processome is a metastable intermediate that ultimately progresses towards a mature SSU through the release of biogenesis factors. The decision to progress the SSU Processome is thought to be an early checkpoint in the SSU assembly pathway, but insight into the mechanisms of progression is needed. Previous studies suggest that Bud23 plays an uncharacterized role during SSU Processome progression. Here, we used a genetic approach to understand its function and found that Bud23 is connected to a network of SSU Processome factors that stabilize the particle. Interestingly, two of these factors are enzymes that are needed for progression. We conclude that Bud23 promotes the release of factors surrounding its binding site to induce structural rearrangements during the progression of the SSU Processome.

## Introduction

Ribosomes are the molecular machines that translate the genetic code. Each ribosome is composed of a small subunit (SSU) that coordinates mRNAs and tRNAs for decoding and a large subunit (LSU) that catalyzes peptide bond formation. In the eukaryotic model organism *Saccharomyces cerevisiae*, the LSU, or 60S subunit, contains three rRNAs (25S, 5.8S and 5S) and 46 ribosomal proteins (r-proteins), whereas the SSU, or 40S subunit, is composed of 18S rRNA and 33 r-proteins [1]. The subunits are produced by an energetically expensive and dynamic assembly pathway requiring more than 200 *trans*-acting biogenesis factors (reviewed in [2–5]). Ribosome assembly begins in the nucleolus with the synthesis of the primary 35S rRNA transcript and the 5S rRNA. The primary transcript contains the 18S, 5.8S, and 25S rRNAs and four spacer regions that are removed during ribosome assembly: two external transcribed spacers (ETS) and two internal transcribed spacers (ITS) (S1 Fig). The r-proteins and the biogenesis factors assemble on the rRNA in a hierarchical order [6–10]. While most of the biogenesis factors promote the correct architecture of the subunits by chaperoning and modifying the rRNA, others drive structural rearrangements and removal of the spacer regions.

Because the pre-18S rRNA is encoded in the 5'-portion of the primary transcript the initial folding of the pre-rRNA is dedicated to SSU assembly. These co-transcriptional nucleolar events lead to assembly of the SSU Processome (sometimes referred to as a 90S pre-ribosome) [8,11,12]. The SSU Processome appears to serve two primary roles: 1) promote the formation of an early, stable precursor of the SSU and 2) process the pre-rRNA at sites A0 and A1, to remove the 5' ETS, and at site A2, to separate the SSU and LSU precursors (reviewed in [13,14]). Recent structures of the complete SSU Processome revealed a large metastable assemblage of 15 r-proteins and about 50 biogenesis factors on the 5' ETS and pre-18S rRNAs [15–17]. The 18S rRNA can be divided into the 5', central, 3' major, and 3' minor domains (S2 Fig) that fold independently as the SSU Processome assembles [15–19]. In the SSU Processome, these RNA domains are scaffolded by a multitude of biogenesis factors, the 5' ETS ribonucleoprotein complex (RNP), and U3 snoRNA that prevent their collapse into the densely packed structure of the subsequent pre-40S and mature 40S particles. A correctly assembled SSU Processome ultimately transitions to the pre-40S [20] which requires the release of most of these biogenesis factors, including U3 and 5' ETS RNP, as well as the aforementioned endonucleolytic cleavages of the pre-rRNA. This transition, involving the coordinated disassembly of the SSU Processome, allows large architectural rearrangements to occur that yield the pre-40S and likely release it into the nucleoplasm for continued maturation. While recent structural and molecular analysis of the SSU Processome have brought its assembly into focus, there remains a dearth of mechanistic understanding of the events driving its transition into the pre-40S.

Early recruitment of the U3 snoRNA is crucial for the formation and function of the SSU Processome [7,11,21–26]. Structural analyses of the SSU Processome show that the box C/D U3 snoRNA threads into the core of the complex where it spatially separates the rRNA domains and scaffolds biogenesis factors [15,16]. Two regions of U3, referred to as the 5' and 3' hinges, hybridize to the 5' ETS RNA while its Box A' and Box A regions hybridize to the pre-18S rRNA, henceforth U3-18S heteroduplexes. Notably, Box A hybridizes with helix 1 of 18S, precluding the formation of the central pseudoknot (CPK) of the SSU [15,17]. The CPK is a universally conserved feature of the SSU formed by long-range, non-canonical base-pairing between helices 1 and 2 that allows the four rRNA domains to compact onto one another (S2 Fig), and generates the environment necessary to establish the decoding center [1,27]. Because U3 blocks CPK formation, the release of U3 is a critical, irreversible step in the maturation of the SSU [14,28]. The unwinding of U3 is catalyzed by the DEAH/RHA helicase Dhr1 [29] which is activated by the SSU Processome factor Utp14 [30–32]. Mutational analysis identified a short loop of Utp14 that is necessary and sufficient for the activation of Dhr1 *in vitro* [30,32], and deletion or mutation in this loop phenocopies a catalytic *dhr1* mutant *in vivo* [30]. How Utp14 times the activation of Dhr1 remains unknown, and the activation loop of Utp14 has not been resolved in SSU Processome structures [15,16]. RNA crosslinking and structural analysis indicate that Utp14 binds simultaneously to the U3 and 5' ETS RNAs, as well as the central domain and 5'- and 3'-ends of the pre-18S rRNA, suggesting that Utp14 is uniquely positioned to time Dhr1 activation by monitoring completion of transcription of the pre-18S rRNA [15,16,33].

The endonucleolytic cleavages within the rRNA at sites A1 and A2 are also irreversible steps that occur around the time of the transition of the SSU Processome to pre-40S (S1 Fig). The complete SSU Processome structures all contain rRNA cleaved at A0 but not at A1, indicating that A0 cleavage alone is does not trigger progression from the SSU Processome [15–17,19]. Around the time of Dhr1 function, cleavages at sites A1 within 5' ETS and A2 within ITS1 occur [29,34,35]. It is possible that cleavage at site A1 sets the transition in motion. Site A1 is cleaved by the PIN domain nuclease Utp24 [36–38]. However, it is positioned about 50 Å away from site A1 in the complete SSU Processome [15,16] indicating that some structural rearrangements must occur for it to access its substrate. Subsequent cleavage at site A2 separates the SSU precursor from the LSU precursor. When cleavage at site A2 is inhibited, cleavage at the downstream site A3 instead bifurcates the two maturation pathways.

Bud23 is a methyltransferase that acts with its cofactor Trm112 to modify guanosine 1575 (G1575) within the 3' major domain of 18S rRNA [39–42]. *BUD23* is a nonessential gene in yeast, but its deletion (*bud23Δ*) causes a substantial growth defect that correlates with an approximate 70% reduction of 40S subunits [39]. Catalytically inactive *bud23* mutants fully complement the growth defect of *bud23Δ* cells suggesting that it is the presence of the protein–but not its methyltransferase activity–that is needed for ribosome assembly [39,40,43]. Bud23 is not a stable component of the complete SSU Processome and is usually thought to act on a nuclear pre-40S intermediate. Consistent with this notion, the human orthologs of Bud23 and Trm112 have been resolved in an early pre-40S structure [44]. Moreover, its binding site is occupied by the assembly factor Emg1 in the SSU Processome structures [15–17], precluding Bud23 from assembling into this structure. Despite these observations, there are multiple lines of evidence indicating that Bud23 joins the SSU assembly pathway during the transition of the SSU Processome to pre-40S. First, Bud23 coimmunoprecipitates with the late-acting SSU Processome factors Dhr1, Utp14, and Utp2 [45,46], and Trm112 copurifies several late SSU Processome factors including Dhr1 and Utp14 [42]. Second, Bud23 and Trm112 sediment at the positions of both 90S and 40S in sucrose density gradients, reflecting their association with the SSU Processome and pre-40S, respectively [40,42,46]. Third, *bud23Δ* cells are defective in A2

site cleavage [45] which releases the pre-40S particle. Finally, extragenic mutations in the SSU Processome factors *DHR1*, *UTP14*, *UTP2 (NOP14)*, and *IMP4* [30,45–47] alleviate the growth and A2 site cleavage defects of *bud23Δ* suggesting that Bud23 acts concurrently with these factors. Despite the evidence that Bud23 enters the 40S biogenesis pathway during the transition of the SSU Processome to pre-40S, the specific function for Bud23 has not been described.

Here, we have carried out a comprehensive genetic analysis of extragenic suppressor mutations of *bud23Δ* and identified a genetic and physical interaction network that connects Bud23 to the transition of the SSU Processome to the pre-40S. We found novel extragenic mutations in *IMP4*, *RPS28A*, *UTP2*, *UTP14*, *DHR1*, and *BMS1* that acted as bypass suppressors of *bud23Δ*. Recent structures provide the context to rationalize how many of these amino acid substitutions disrupt SSU Processome structure and suggest how Bud23 works in the SSU Processome transition. Bms1, Imp4, and Utp2 all interact with the 3' major domain and have extensions that embrace the U3-18S substrate of Dhr1. We found that many of the substitutions destabilized physical interactions within this network and genetically connect the 3' major domain to the U3-18S heteroduplexes. Finally, mass spectrometric and Northern blot analysis of particles isolated in the absence of Bud23 revealed an enrichment of late-acting SSU Processome factors and rRNA species with defective processing. Together, our data imply that Bud23 binding induces the disassembly of SSU Processome factors connecting the 3' major domain to the U3-18S duplexes. We propose that Bud23 promotes rearrangements of the 3' major domain to drive Bms1 and Dhr1 function to generate the pre-40S intermediate.

## Results

### Extragenic suppressors of *bud23Δ* map to SSU Processome factors and connect Bud23 to the U3 snoRNA

Bud23 methylates G1575 of the 18S rRNA [39,40]. This residue is located in the lower region of the 3' major domain, which comprises helices 28, 29, 30 41, 41es10, 42, and 43 and termed the 3' basal subdomain (Fig 1A and 1B, and S2 Fig) [17]. The deletion of *BUD23* severely impairs 40S production and cell growth, yet a catalytically inactive Bud23 fully complements *bud23Δ* [39,40], suggesting that 40S assembly requires Bud23 binding but not rRNA methylation. The slow-growth defect of *bud23Δ* places strong selective pressure on cells for extragenic bypass suppressors. Our lab previously reported suppressing mutations in *DHR1*, *UTP14*, and *UTP2* which code for late-acting SSU Processome factors [30,45,46]. We also found mutations in *IMP4* encoding an early SSU Processome factor [47]. These results connected Bud23 to the late events of the SSU Processome, but they did not allow us to rationalize a mechanism for Bud23 function. The complete SSU Processome harbors nearly 70 ribosomal proteins and biogenesis factors, and we postulated that mapping the mutated residues to structures of the SSU Processome would help illuminate the function of Bud23. To expand the coverage, we screened for additional spontaneous suppressors by continuously passaging cultures of *bud23Δ* until they arose. We then amplified and sequenced the *IMP4*, *DHR1*, *UTP14*, and *UTP2* loci from these suppressed strains, and identified additional mutations in these genes. Suppressed strains that did not contain mutations in these genes were subjected to whole-genome sequencing and genome variant analysis. This revealed novel suppressing mutations in *RPS28A*, a ribosomal protein that binds the 3' basal subdomain, and *BMS1*, an essential GTPase of the SSU Processome. Mutations in *RPS28A* and *BMS1* were confirmed by Sanger sequencing and verified as suppressors by reintroducing them on vectors into *bud23Δ* cells.

Portions of Bms1, Imp4, Rps28, Utp2, and Utp14, but not Dhr1, have been resolved in structures of the complete SSU Processome [15–17]. Remarkably, Bms1, Imp4, Rps28, and Utp2 all interact directly with the 3' basal subdomain which contains the Bud23 binding site

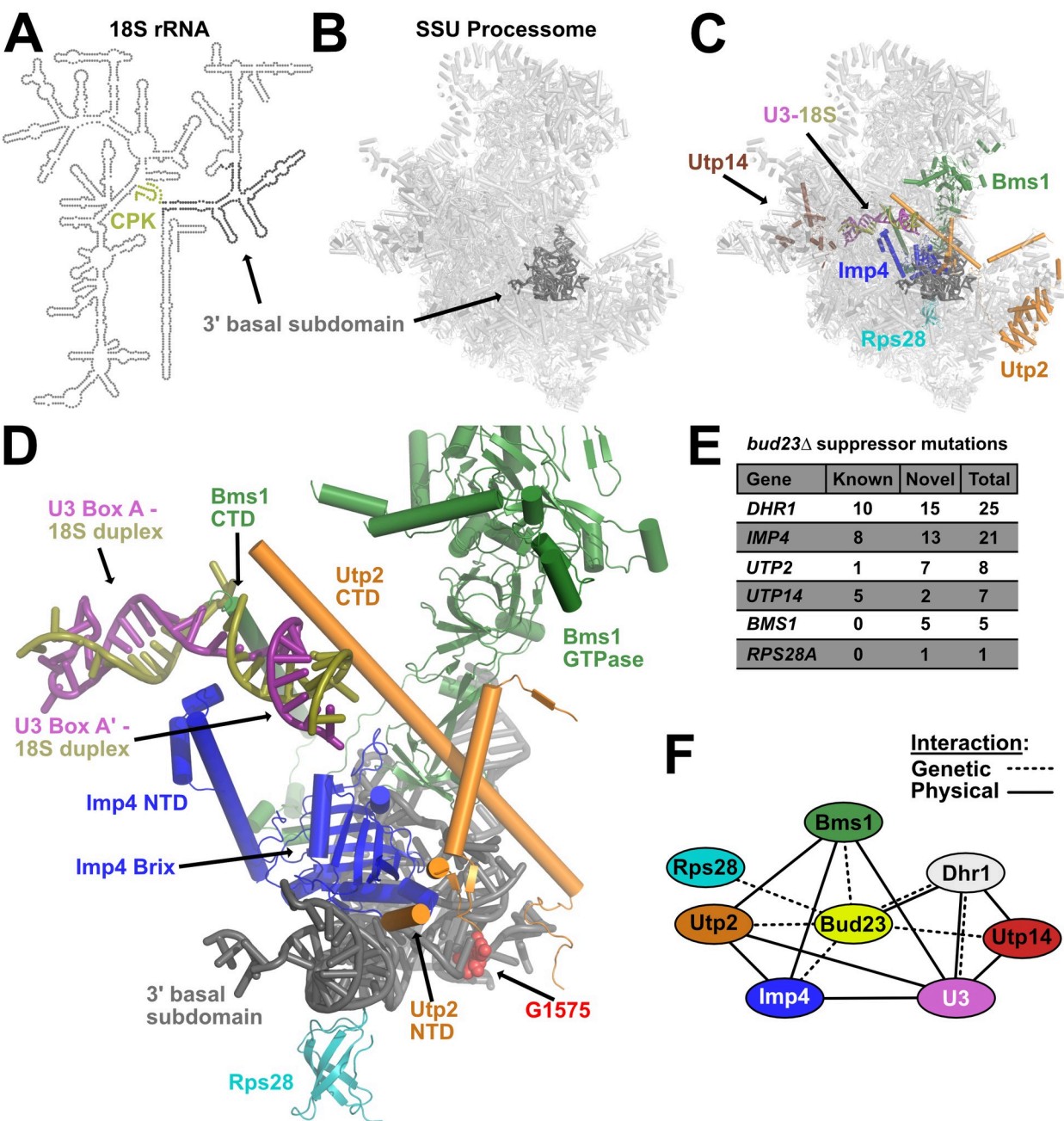

**Fig 1. Extragenic suppressors of *bud23Δ* reveal an interaction network that connects the 3' basal subdomain to the U3-18S heteroduplexes.** (A) A secondary structure map of the 18S rRNA that indicates the position of the 3' basal subdomain (dark gray) and the central pseudoknot (CPK; deep olive). (B) The position of the 3' basal subdomain (dark gray) within the context of the SSU Processome structure (light gray). (C) Factors harboring mutations that suppress *bud23Δ* and are resolved in the SSU Processome (PDB 5WLC) cluster around the 3' basal subdomain and the U3-Box A'-18S heteroduplex that Dhr1 unwinds. Shown are: Bms1 (forest green), Imp4 (blue), Rps28 (cyan), Utp2 (orange), Utp14 (brown), U3 (deep purple), 18S rRNA (deep olive), 3' basal subdomain (dark gray). (D) Zoomed view of factors in C showing contacts amongst each other, the 3' basal subdomain, and the U3-Box A'-18S heteroduplex. N- and C-terminal domains, NTD and CTD, respectively. The U3-18S heteroduplexes are shown as U3 Box A and U3 Box A'. Guanosine 1575 (G1575, red) is shown as a marker for the binding site of Bud23. (E) Tabulation of the number of unique mutations found in each extragenic suppressor of *bud23Δ*. Newly identified mutations (novel) and previously identified (known) [30,45–47]. The complete list of these mutations is available in S1 Table. (F) Summary of the genetic and physical interactions amongst the suppressors of *bud23Δ*. Factors are indicated as nodes; genetic and physical interactions are shown as dashed and solid edges, respectively.

(Fig 1C). These factors appear to contribute to the structural stability of the SSU Processome and form multiple protein-protein and protein-RNA contacts (Fig 1D). Additionally, Bms1, Imp4, and Utp2 each contain extended alpha-helices that penetrate into the core of the SSU Processome where they embrace the U3 Box A and Box A'-18S duplexes (Fig 1D). We previously determined that Dhr1 binds to the 5'-hinge and Box A of U3 which is located immediately upstream of and overlapping the Box A and Box A'-18S duplexes that we identified as its substrate (S3A Fig) [29]. Only a few segments of Utp14 are resolved in current SSU Processome structures, but Utp14 can be seen binding to pre-rRNA and U3 snoRNA immediately upstream of the duplexes Dhr1 unwinds (S3A Fig) [15,33]. Intriguingly, Utp14 and the factors positioned at the 3' basal subdomain bookend the U3-18S heteroduplexes. Thus, Imp4, Utp2, and Bms1 provide a physical linkage between the 3' basal subdomain and the U3-18S heteroduplexes that are unwound by Dhr1.

We identified five novel mutations in *BMS1* and one in *RPS28A* as spontaneous suppressors of *bud23Δ* (Fig 1E and S1 Table). We also found an additional 15 mutations in *DHR1*, 13 mutations in *IMP4*, two mutations in *UTP14*, and one mutation in *UTP2* that were not isolated in our previous studies. Five additional mutations were identified in *UTP2* using error-prone PCR mutagenesis (discussed below). These observations revealed a network of SSU Processome factors that genetically interact with Bud23 and make multiple physical contacts amongst one another (Fig 1F). Importantly, this interaction network physically connects the 3' basal subdomain with the U3-18S heteroduplex substrates of Dhr1, suggesting a functional linkage between these two sites. Many of the amino acid substitutions that we report here and previously [30] are in protein-RNA or protein-protein interfaces where they would appear to weaken interactions within the SSU Processome. Because these mutations bypass the absence of Bud23, we propose that Bud23 binding to the 3' basal subdomain induces the release of factors from this region to promote progression of the SSU Processome to a pre-40S particle. In the following sections, we consider how the *bud23Δ* bypass suppressors affect the dynamics of the particle.

## The amino acid changes in Imp4 and Rps28A mainly cluster around their interfaces with the 3' basal subdomain

We identified 21 unique mutations in *IMP4* and a single mutation within *RPS28A* that suppressed *bud23Δ* (Fig 1E). All of these mutations partially restored growth in a *bud23Δ* mutant (Fig 2A), although the *rps28A-G24D* mutation did not suppress as well as the *imp4* mutations, perhaps because expression of the wild-type paralog *RPS28B* partially masked its suppression phenotype. Imp4 is a component of the heterotrimeric Mpp10-Imp3-Imp4 sub-complex [48–52] which enters the SSU Processome at an early stage of its assembly, during transcription of the 5' ETS [6,7]. The Mpp10 complex may serve as an initial binding platform for several additional SSU Processome factors [53]. Imp4 is positioned in the core of the SSU Processome where its N-terminal domain (NTD) contacts the U3-18S heteroduplexes while its RNA-binding Brix domain is cradled in the concave RNA fold of the 3' basal subdomain (Fig 2B) [15–17]. On the other hand, the ribosomal protein Rps28 binds to the opposite, convex surface of the 3' basal subdomain (Fig 2B), adjacent to but not occluding the Bud23 binding site (Fig 2B; marked by G1575) [44].

The amino acid substitutions in Imp4 primarily mapped to two regions of the protein (Fig 2C). Most of the substitutions were in the Brix domain of Imp4, at its interface with the 3' basal subdomain RNA, henceforth "rRNA interaction" substitutions (Fig 2C). These included substitutions of residues S93, R94, S101, R116, N118, N121, and R146 which are all expected to form hydrogen bonds with the rRNA [15,16]. These substitutions likely weaken the affinity of

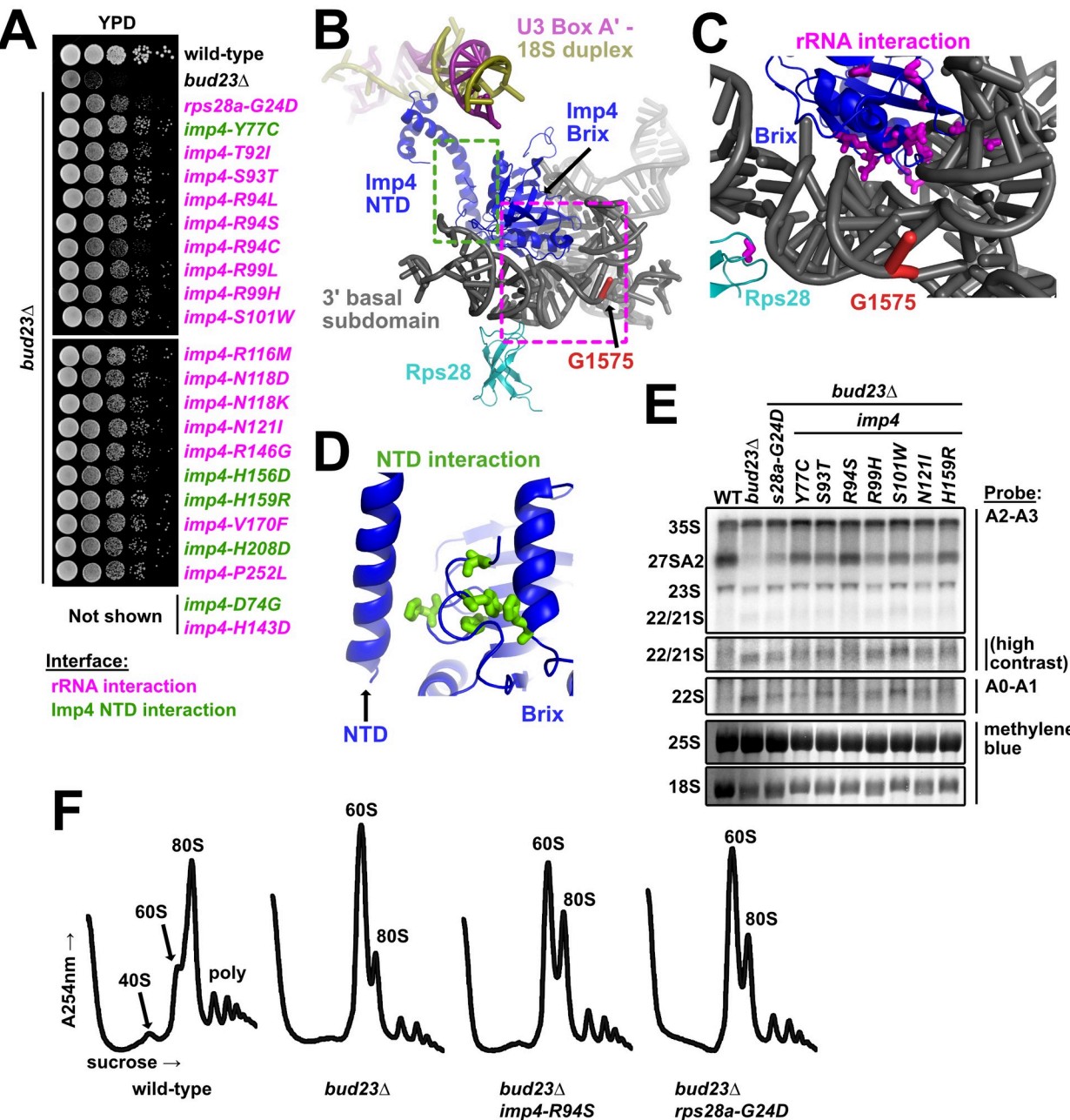

**Fig 2. The mutated residues in Imp4 and Rps28A primarily map to their interfaces with the 3' basal subdomain.** (A) Point mutations within *imp4* and *rps28a* suppressed the growth defect of *bud23Δ* as shown by 10-fold serial dilutions of wild-type cells (BY4741), *bud23Δ* (AJY2676), and *bud23Δ*-suppressed cells spotted on YPD media and grown for two days at 30°C. (B) Rps28 and the Brix domain of Imp4 interact with the 3' basal subdomain RNA, while the NTD of Imp4 makes contacts with its Brix domain and the U3-18S heteroduplexes. G1575, the binding site of Bud23, is shown for reference. The regions where the mutated residues map are indicated by magenta and green dashed boxes for the rRNA interaction and the NTD interaction, respectively. Factors are colored the same as in Fig 1. (C) Residues mutated in Rps28 and Imp4 Brix domain map to interaction interfaces with the 3' basal subdomain RNA (magenta sticks). (D) Several residues mutated in Imp4 map to an intramolecular interaction between the Brix and NTD of Imp4 (green sticks). (E) Suppressing mutations in *imp4* and *rps28a* partially restored A2 processing and 18S rRNA levels in *bud23Δ* cells. RNA processing intermediates were detected by Northern blotting on RNAs extracted from wild-type (WT), *bud23Δ*, and *bud23Δ*-suppressed cells cultured to exponential phase at 30°C in liquid YPD. P32-radiolabeled probes (Table 3) hybridized to the indicated regions. The 25S and 18S rRNAs were detected by methylene blue staining of the RNAs prior to oligonucleotide hybridization. (F) The *imp4* and *rps28a* mutations partially restored 40S biogenesis as shown by polysome profiles after separation of extracts on sucrose density gradients from wild-type, *bud23Δ*, and *bud23Δ*-suppressed cells cultured to exponential phase at 30°C in liquid YPD media.

the protein for the rRNA. The single Rps28-G24D substitution maps to its rRNA interface with the 3' basal subdomain. Unlike Imp4, Rps28 is an integral component of the small subunit and remains associated with the mature ribosome. Consequently, it is unlikely that the glycine to aspartate substitution promotes release of Rps28. More likely, this substitution may increase the flexibility of the RNA to facilitate release of Imp4 (Fig 2C). Five of the substitutions in Imp4, in residues D74, Y77, H156, H159 and H208, mapped to an intramolecular domain interface between the core of the protein and the NTD that interacts with the U3 Box A'-18S duplex, henceforth "NTD interaction" substitutions (Fig 2D). The NTD interaction substitutions may alter the flexibility of the NTD, thereby destabilizing its interaction with the U3 Box A'-18S duplex. The observation that *imp4* mutations that suppress *bud23Δ* are predicted to weaken the affinity of Imp4 for the 3' basal subdomain, suggests that Bud23 binding to the 3' basal subdomain leads to disruption of the protein-RNA interactions in this region.

The individual suppressing mutations complemented as well as wild-type (WT) *IMP4* (S4A Fig). To generate a mutant with a stronger phenotype to facilitate molecular analysis, we focused on the interaction between Imp4 and the rRNA of the 3' basal subdomain. We posited that combining Imp4 substitutions within this interface would further destabilize the interaction and lead to measurable molecular defects. To this end, we generated three additional *imp4* mutants with combinations of mutations: containing both S93T and R94S (*imp4-S93T, R94S*), R116M and N118D (*imp4-R116M, N118D*), or all four mutations (*imp4-TSMD*). Both *imp4* double mutants partially complemented the loss of *IMP4* while *imp4-TSMD* did not (S4A Fig), consistent with increased loss of function as mutations were combined. Similarly, the two *imp4* double mutants were weaker suppressors of *bud23Δ* than their constituent single mutants while *imp4-TSMD* was unable to suppress *bud23Δ* (S4B Fig). Surprisingly, *imp4-TSMD* was strongly dominant negative, suggesting that although it was non-functional, it retained interaction with binding partners and may assemble into the SSU Processome. To ask whether Imp4-TSMD retained association with pre-ribosomal particles, we first generated 13xMYC tagged wild-type Imp4 and Imp4-TSMD constructs. Ectopic expression of WT *IMP4-13xMYC* complemented as well untagged *IMP4* (S4C Fig; left panel) indicating that the tagged protein was fully functional. We then monitored the sedimentation of the tagged Imp4 proteins in sucrose density gradients (S4C Fig; right panel). Imp4-TSMD-13xMYC sedimented throughout the gradient, similar to the behavior of WT Imp4-13xMYC, suggesting association with pre-ribosomes. We interpret these results to suggest that the substitution of residues within the Imp4-rRNA interface destabilize the local environment, likely disrupting proper SSU Processome function, but they do not prevent Imp4 from binding pre-ribosomes because of its extensive contacts with multiple SSU Processome factors [15–17,48,51–55]. This interpretation is consistent with the dominant negative effect of the *imp4-TSMD* mutant. We conclude that single amino acid changes in Imp4 partially bypass the absence of Bud23 by subtly destabilizing specific interfaces within the SSU Processome, but on their own, do not perturb SSU Processome function to a degree that results in a growth defect.

Bud23 is needed for efficient A2 site processing [39]. To ask if the suppressing mutations in *IMP4* and *RPS28A* bypass this rRNA processing defect in *bud23Δ* cells, we prepared total RNA from actively dividing wild-type (WT) cells or *bud23Δ* cells with or without a suppressing mutation in *IMP4* or *RPS28A* and probed for rRNA processing intermediates by Northern blotting (Fig 2E). As we reported previously [45], *bud23Δ* cells showed a loss of the 27SA2 rRNA intermediate, indicating loss of A2 cleavage, and reduced levels of 18S rRNA compared to WT cells, but no concurrent accumulation of 23S rRNAs. Suppression of *bud23Δ* by the *imp4* and *rps28A* mutants partially restored levels of the 27SA2 rRNA intermediate and 18S rRNA indicating a restoration of cleavage at site A2 and 40S biogenesis. Surprisingly, *bud23Δ* cells also slightly accumulated the 22/21S intermediates (Fig 2E). 22S represents rRNA cleaved

at sites A0 and A3 but not A1 or A2 while 21S represents rRNA cleaved at sites A1 and A3 but not A2. (S1 Fig). Although the A2-A3 probe cannot distinguish between the 22S and 21S intermediates the A0-A1 probe gave a similar hybridization signal indicating that the 22S rRNA accounts for some of this signal (Fig 2E). Suppression of *bud23Δ* partially alleviated the accumulation of this species. This was most evident in the strains harboring the *imp4* mutants S93T, R94S, N121I, and H159R. Although these data indicate that Bud23 affects not only A2 processing, as we previously reported [39], but also cleavage at A1, we suspect that the effect on A1 cleavage is indirect.

As a complementary approach to ask if *bud23Δ* suppressors restored 40S biogenesis, we analyzed ribosomal subunit levels on sucrose density gradients after separating free ribosomal subunits, 80S, and polysomes from wild-type cells and *bud23Δ* cells with or without a suppressing mutation. In wild-type cells, there was an appreciable steady-state level of free 40S and 60S subunits (Fig 2F). In contrast, in *bud23Δ* cells in which 40S production is limited, the free 40S peak disappeared and the amount of free 60S was dramatically increased, at the expense of 80S (Fig 2F). The introduction of suppressing mutations in *imp4* or *rps28a* partially restored the levels 80S and free 40S, similar to the suppression of *bud23Δ* by mutations in *utp14* or *utp2* that we reported previously [45]. Taken together, the Northern blotting and sucrose density gradient data indicate that *imp4* and *rps28A* mutants partially alleviate the 40S biogenesis defects of *bud23Δ* cells.

## The Utp2 mutants destabilize its interaction with Imp4

Our lab previously identified *utp2-A2D* as a spontaneous and dominant suppressor of *bud23Δ* that partially restores 40S biogenesis and A2 processing of the primary rRNA transcript [45]. From our screen for additional spontaneous suppressors of *bud23Δ*, we found an additional *UTP2* mutation, *utp2-L9S*, that suppressed *bud23Δ* (Fig 3A). Utp2, also known as Nop14, assembles into the SSU Processome with its binding partners Emg1, Noc4, and Utp14 [54–56], once the 3' minor domain is fully transcribed [6,7]. The association of human Utp2 with human pre-40S complexes indicates that Utp2 remains on nascent particles during the transition from the SSU Processome to pre-40S [44] suggesting that it has an active role in particle progression.

To gain further insight into the mechanism by which mutations in *UTP2* suppress *bud23Δ*, we performed random PCR mutagenesis of the entire *UTP2* coding sequence and identified six additional mutations in *UTP2* that suppressed *bud23Δ* to different degrees (Fig 3B; left panel). In this screen we also reisolated the previously identified *utp2-A2D* mutation. All mutants fully complemented loss of Utp2 (Fig 3B; right panel). The suppressing mutations all mapped to the N-terminal domain of Utp2; four clustered around the extreme N-terminus and another three clustered around residues 148–151. In an attempt to generate mutants with stronger phenotypes than the individual mutants, we generated the combinatorial mutants *utp2-DPE* containing the mutations A2D, L6P, and K7E, and *utp2-SSH* harboring the mutations L148S, F149S, and L151H. Both of the combinatorial mutants retained the ability to suppress *bud23Δ* and fully complemented loss of *UTP2*, but *utp2-SSH* was a stronger suppressor than *utp2-DPE* (Fig 3B).

Based on recent partial structures of Utp2 within the SSU Processome [15,16] the globular domain of Utp2 directly contacts the Emg1 heterodimer, Enp1, and Noc4 within a region of the 3' major domain that will make up the beak of the mature SSU, while its extended N- and C-terminal arms extend over the 3' basal subdomain and pierce into the core of the SSU Processome. The C-terminal arm of Utp2 contacts the U3 Box A'-18S duplex while its NTD contacts the Brix domain of Imp4 (Figs 1D and 3C). Notably, four of the residues (L6, K7, L9 and

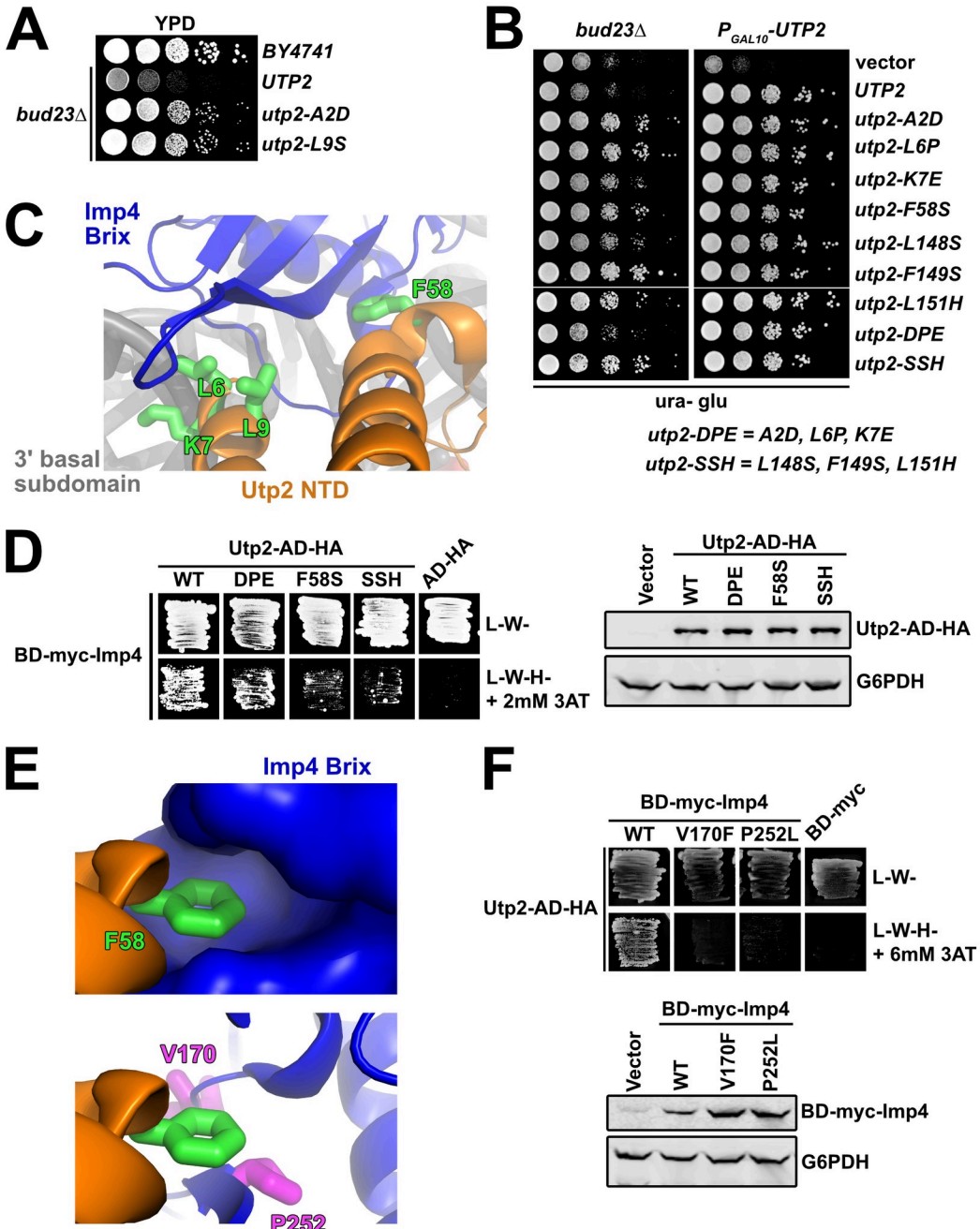

**Fig 3. The Utp2 mutants lose interaction with Imp4.** (A) Spontaneous point mutations within *utp2* partially suppressed the growth defect of *bud23Δ* as shown by 10-fold serial dilutions of wild-type cells (BY4741), *bud23Δ* (AJY2676), and *bud23Δ*-suppressed cells spotted on YPD media and grown for two days at 30˚C. (B) Additional point mutations in *UTP2*, generated by error-prone PCR, also suppressed the growth defect of *bud23Δ* (left) and complemented loss of *UTP2* (right) as shown by 10-fold serial dilutions of *bud23Δ* (AJY2676) and *PGAL10-UTP2* (AJY4175) cells containing either empty vector (pRS416), or vectors encoding the indicated alleles of *UTP2* spotted on SD-Ura media containing glucose and grown for two days at 30˚C. (C) Several of the amino acid substitutions in Utp2 map to residues (green sticks) located within its NTD (orange) that interacts with the Brix domain of Imp4 (blue) adjacent to the 3' basal subdomain RNA (gray). (D) Left: Yeast two-hybrid interaction assay between Imp4 and wild-type (WT) or mutant Utp2. Strains carrying the indicated constructs were patched onto SD-Leu-Trp- (L-W-) and SD-Leu-Trp-His- (L-W-H-) media supplemented with 2 mM 3-Amino-1,2,4-triazole (3AT) (AD, Gal4 activation domain; BD, Gal4 DNA binding domain). Right: Western blot analysis of the wild-type and mutant Utp2-AD-HA proteins using equivalent amounts of total protein extracts. Glucose-6-phosphate dehydrogenase (G6PDH) was used as a loading control. (E) Top: F58 of Utp2 (green sticks) fits into a hydrophobic pocket in the Brix domain of Imp4 (surface representation). Bottom: The *bud23Δ*-suppressing mutations

V170F and P252L of Imp4 (magenta sticks) line this pocket. Imp4 and Utp2 are colored blue and orange, respectively. (F) Top: Yeast two-hybrid interaction assay between Utp2 and wild-type or mutant Imp4. Strains carrying the indicated constructs were patched onto L-W- and L-W-H- media supplemented with 6 mM 3AT. Bottom: Western blot analysis of the wild-type and mutant BD-myc-Imp4 proteins in equivalent amounts of total protein extract is shown. G6PDH was used as a loading control.

F58) mutated in our screen were resolved in the SSU Processome structures (Fig 3C). Residues L6, K7, and L9 are within a small helix on the extreme N-terminus of Utp2 that interacts with Imp4, while K7 also appears to contact the phosphate backbone of C1623 of the 3' basal subdomain. Meanwhile, F58 of Utp2 makes an additional nearby contact between these proteins. These observations prompted us to speculate that the *utp2* suppressors of *bud23Δ* perturb the interaction between Utp2 and Imp4. Previous large-scale yeast-two hybrid (Y2H) studies did not report an interaction between Utp2 and Imp4 [54,55]. However, those studies used Utp2 constructs harboring N-terminal fusions of GAL4 activating or DNA binding domain (AD and BD, respectively). Because the apparent interaction between Utp2 and Imp4 requires the extreme N-terminus of Utp2 (Fig 3C), such a fusion protein could sterically hinder their interaction. To this end, we cloned a Utp2 Y2H construct harboring an HA-tagged GAL4 activating domain fused to its C-terminus (Utp2-AD-HA), which allowed us to detect an interaction between Utp2-AD-HA and BD-myc-Imp4 (Fig 3D; left panel). Using this system, we assayed the Utp2-DPE, Utp2-F58S, and Utp2-SSH mutants for their ability to interact with Imp4. All of the mutants showed decreased interaction with Imp4 with the Utp2-F58S and Utp2-SSH mutants being the most severe. All the mutant Utp2 proteins were expressed to similar levels indicating that the reduced interaction was not due to differences in expression or degradation of the mutant proteins (Fig 3D; right panel). The results from the analysis of Utp2-DPE and Utp2-F58S are consistent with the notion that the mutations in *UTP2* that suppress *bud23Δ* disrupt the interaction between Utp2 and Imp4. The result of Utp2-SSH losing interaction with BD-Imp4 suggests that the flexible, unresolved region of Utp2 between R116 and P201 interacts with Imp4, but we cannot rule out the formal possibility that these mutations suppress *bud23Δ* by some other means.

F58 of Utp2 fits into a hydrophobic pocket of Imp4 (Fig 3E; upper panel). Interestingly, two mutations in *IMP4*, V170F and P252L, that suppressed *bud23Δ* (Fig 2A and 2B) map to residues that line this pocket (Fig 3E; lower panel). The positions of these two altered residues predict that they could also disrupt the interaction between Imp4 and Utp2. To test this possibility, we introduced V170F and P252L into the BD-myc-Imp4 vector and assayed these mutants for their interaction with Utp2-AD-HA. Indeed, substitution of either residue caused a loss of interaction between Utp2 and Imp4 (Fig 3F; upper panel), and the loss of interaction could not be explained by reduced protein expression of the mutant Imp4 constructs (Fig 3F; lower panel). These results, together with the Y2H assays using mutant Utp2 strongly suggest that disrupting the interaction between Imp4 and Utp2 bypasses the 40S assembly defect in the absence of Bud23.

## The Bms1 mutants are poised to affect its conformational state

We found five *bud23Δ*-suppressing mutations within *BMS1* (Figs 1F and 4A). Like the mutations in *DHR1*, *UTP14*, *UTP2*, *IMP4*, and *RPS28A* (Fig 2E) [30,45,46], mutant *bms1* alleles partially alleviated the rRNA processing defects and restored 40S biogenesis (Fig 4B and 4C) suggesting that these mutations overcome the same biogenesis defect in the absence of Bud23 that the other *bud23Δ* suppressors do. The mutant *bms1* alleles complemented the loss of endogenous *BMS1* as well as wild-type *BMS1* in *BUD23*-replete cells (S5 Fig) indicating that

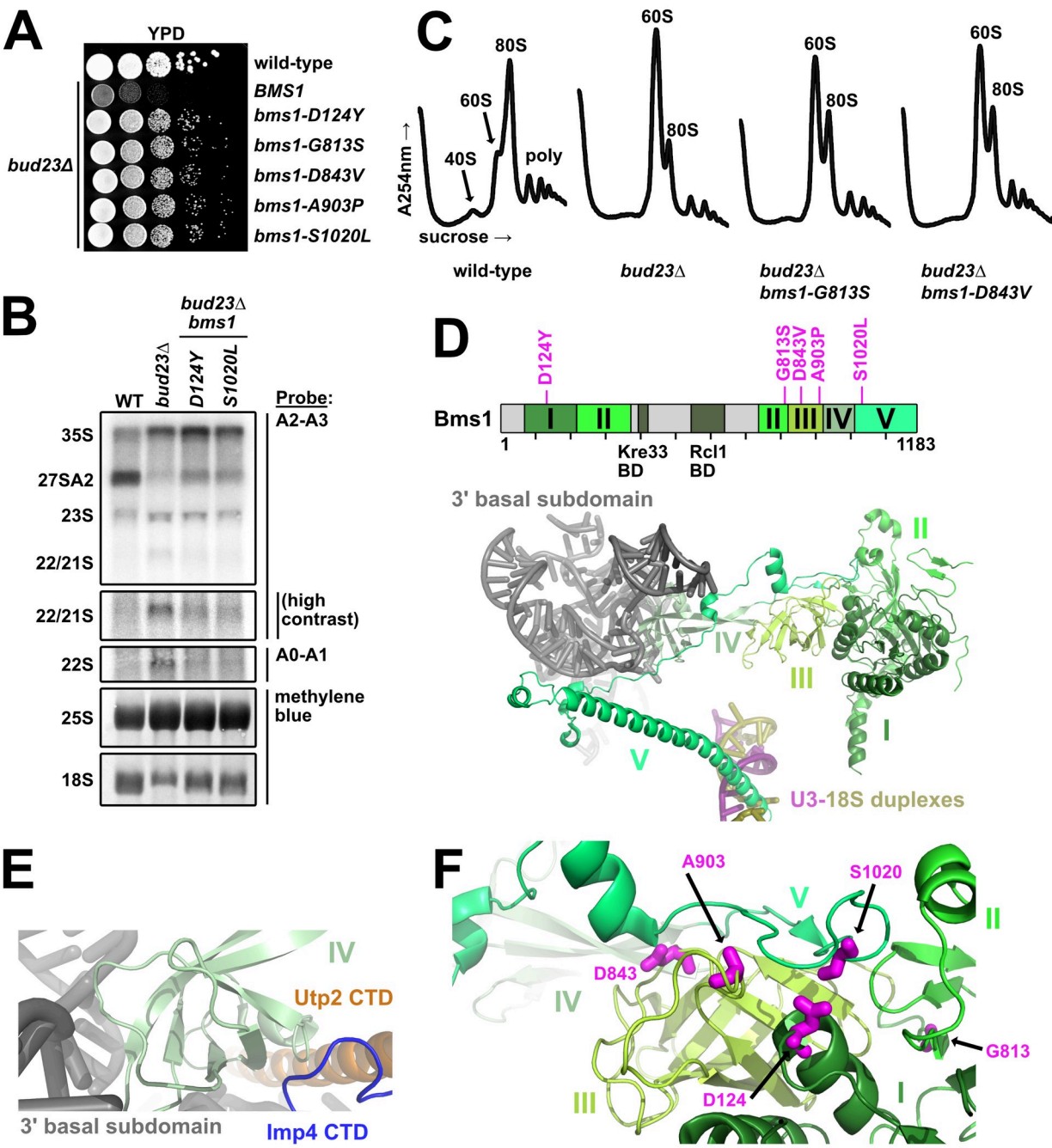

**Fig 4. The mutated residues in the GTPase Bms1 that suppress *bud23Δ* are poised to modulate its conformational state.** (A) Spontaneous point mutations within *BMS1* suppressed the growth defect of *bud23Δ* as shown by 10-fold serial dilutions of wild-type cells (BY4741), *bud23Δ* (AJY2676), and *bud23Δ* cells carrying the indicated *bms1* mutations spotted on YPD media and grown for two days at 30˚C. (B) The *bms1* mutations partially restored A2 processing and 18S rRNA production in *bud23Δ* cells as shown by Northern blotting of RNAs extracted from wild-type, *bud23Δ*, and *bud23Δ*-suppressed cells as described in Fig 2E. (C) The *bms1* mutations partially restored 40S biogenesis as shown by the analysis of the polysome profiles from the indicated strains as described in Fig 2F. (D) Top: Primary structure of Bms1 with domains (in different shades of green), interacting regions and *bud23Δ* suppressing mutations annotated; regions not resolved in SSU Processome structures are indicated in light gray. Bottom: The partial structure of Bms1 (from PDB 5WLC) in the context of the SSU Processome is shown. Domains IV and V extend from its GTPase core (domains I–III) to contact the RNAs of the 3' basal subdomain (gray) and the U3-18S heteroduplexes (pink/gold), respectively. (E) At the 3' basal subdomain, Domain IV of Bms1 also contacts the CTDs of Utp2 (orange) and Imp4 (blue). (F) The mutated residues D124, D843, A903, and S1020 in Bms1 map to inter-domain contacts with the unstructured strand of domain V that connects it to domain IV.

these mutations do not obviously disrupt Bms1 function when Bud23 is present. *BMS1* encodes a 136 kDa GTPase that is essential for 40S biogenesis [57,58]. Bms1 forms a subcomplex with Rcl1 [58–61] which assembles into the SSU Processome once the 3' minor domain is transcribed [6,7]. The GTPase activity of Bms1 has been confirmed *in vitro* [59,60], and its ability to bind GTP is essential [61] suggesting that it is a functional GTPase *in vivo*. GTPases often serve as molecular switches that undergo conformational changes (reviewed in [62]); however, the specific role of Bms1 within the SSU Processome has not been well explored. Due to its position in the SSU Processome, it has been suggested that Bms1 helps remodel the SSU Processome core during the transition to the pre-40S [16].

More than half of Bms1 has been resolved in the SSU Processome structures, and it can be divided into five major domains and two small regions that bind its partner Rcl1 and the acetyltransferase Kre33 (Fig 4D) [15–17]. Domain I contains its catalytic site and, together with the beta-barrels of domains II and III, forms a globular body. Domain IV protrudes from this globular body to interact with the 3' basal subdomain RNA and contacts the CTDs of Imp4 and Utp2 (Fig 4E). Finally, the C-terminal domain V begins as an extended strand that lays on domain III before becoming an extended alpha-helix that inserts between the U3 Box A'-18S and U3 Box A-18S heteroduplexes (Fig 4D). Although the five amino acid substitutions that suppressed *bud23Δ* map to domains I, II, III and V (Fig 4D; upper panel), in 3D structure the mutated residues D124, D843, A903, and S1020 lie directly under the unstructured strand that connects domains IV and V (Fig 4F). Thus, four of the five substitutions likely promote the flexibility of this connecting loop. G813 is located in the connector between Domains II and III where substitution of it could alter the relative positioning of these two domains and influence how domain III interacts with the unstructured strand of domain V.

Bms1 is structurally related to the translation elongation factor EF-Tu which delivers amino-acyl tRNAs to the ribosome (S6A Fig) [57]. Comparison of the Bms1 structure from the SSU Processome to the crystal structures of EF-Tu bound to GDP or the non-hydrolysable GTP analog, GDPNP, suggests that Bms1 is in the GTP-bound state and allows us to speculate how it functions. The beta-barrels of domains II and III of Bms1 are conserved in EF-Tu (S6B Fig). In the GTP-bound state, the beta-barrels of EF-Tu are positioned to accommodate tRNA binding (S6C Fig) [63]. In GDP-bound EF-Tu the beta-barrels are rotated to promote tRNA release (S6C and S6D Fig) [64]. Interestingly, the comparison of the EF-Tu and Bms1 structures revealed that the space occupied by tRNA in EF-Tu (S6E Fig) is occupied by an N-terminal helix of Mpp10 and the unstructured strand of Bms1 that connects domain IV to the extended C-terminal domain V that interacts with U3 (S6F Fig). This observation suggests that the GTP hydrolysis-induced conformational changes of Bms1 could facilitate undocking of the unstructured strand of Bms1 and Mpp10 from the GTPase core of Bms1. Notably, four of the five altered residues in Bms1 that suppressed *bud23Δ* were either within or contact this strand (S6F Fig). We suggest that the substitution of these residues facilitate the release of this strand and, perhaps, Mpp10 from Bms1. Future molecular analysis is needed to better characterize the role of Bms1 in SSU Processome disassembly.

## Disruption of Dhr1 and Utp14 interaction suppresses *bud23Δ*

Our lab previously reported 10 mutations in *DHR1* that suppress *bud23Δ* [46]. Here, we isolated an additional 15 suppressing mutations within *DHR1* (Figs 1F and 5A). Dhr1 is the DEAH-box RNA helicase responsible for unwinding the U3 snoRNA from the SSU Processome [29,34]. The protein harbors a conserved helicase core containing two RecA domains, a Winged-helix (WH) domain, a Helical Bundle (HB) domain, and an OB-fold domain (Fig 5A). Dhr1 also contains an N-terminal domain (NTD) that interacts with Bud23 [40,46] and a

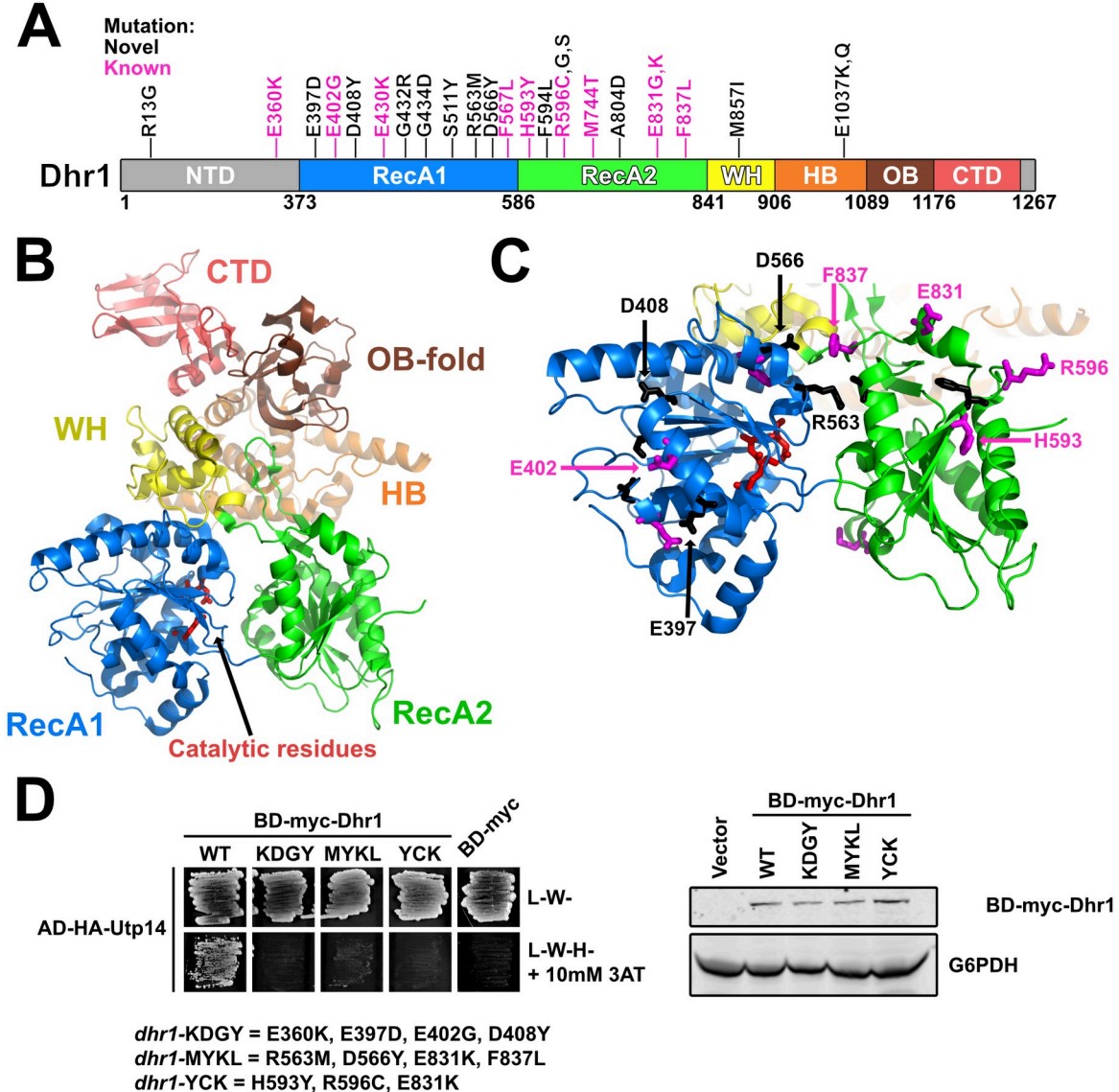

**Fig 5. Most of the *DHR1* mutations map to surface residues of its RecA domains.** (A) A cartoon of the primary structure of Dhr1 is shown. The domains of Dhr1 are annotated by color: NTD, N-terminal domain (light gray); RecA1/2, Recombination protein A1/2 (blue/green); WH, winged-helix (yellow); HB, helical bundle (orange); OB, oligonucleotide-binding fold (brown); CTD, C-terminal domain (light red). Unstructured regions are colored as light gray. Mutations reported here (novel) and previously (known) [46] are indicated as black and magenta, respectively. Numbering indicates residue numbering of yeast Dhr1. (B) The structure of yeast Dhr1 (PDB 6H57) with relevant features colored as described in panel A. Catalytic residues involved in ATP hydrolysis are denoted as red sticks for reference. (C) The majority of the mutated residues map to the surfaces of the RecA domains. Mutated residues are shown as black and magenta sticks as described for panel A. The residues that were used to test loss-of-interaction with Utp14 in panel D are labeled. (D) Top: Yeast two-hybrid interaction data between AD-HA-Utp14 and wild-type (WT) or mutant BD-myc-Dhr1 are shown. Strains carrying the indicated constructs were patched onto SD-Leu-Trp- (L-W-) and SD-Leu-Trp-His- (L-W-H-) media supplemented with 10 mM 3-Amino-1,2,4-triazole (3AT) (AD-HA, GAL4AD-HA; BD-myc, GAL4BD-myc). Bottom: Western blot analysis of the wild-type and mutant BD-myc-Dhr1 proteins using equivalent amounts of total protein extracts is shown. Glucose-6-phosphate dehydrogenase (G6PDH) was used as a loading control.

unique C-terminal domain (CTD) that enhances its interaction with its activator Utp14 [30–32]. Recent crystal structures of recombinant yeast Dhr1 [31] and its murine homolog DHX37 [32] lacking the NTD allow us to map most of the mutated residues to structure (Fig 5B). Consistent with our previous report [46], the overwhelming majority of the substitutions mapped

to residues on the surface of the RecA1 and RecA2 domains (Fig 5C), and fully complemented the loss of *DHR1* (S7 Fig). We previously reported that Utp14, the cofactor of Dhr1, binds the RecA1/2 domains [30]. Based on this, we hypothesized that the amino acid changes within the RecA1/2 domains could affect its interaction with Utp14. To this end, we again turned to Y2H analysis between BD-myc-Dhr1 and AD-HA-Utp14. Our previous analysis of the Dhr1-interacting loop of Utp14 revealed that a combination of substitutions was required in order to observe a loss-of-interaction by Y2H [30]. With this result in mind, we made three Dhr1 Y2H constructs that combined several amino acid changes within or proximal to the RecA1 domain (E360K, E397D, E402G, D408Y), within the RecA2 domain (H593Y, R596C, E831K), or in both the RecA1 and RecA2 domains (R563M, D566Y, E831K, F837L) and tested them for interaction with Utp14. All three mutants showed a loss of interaction with Utp14 (Fig 5D; left). All three of the mutant proteins expressed similarly (Fig 5D; right), indicating that the loss-of-interaction was not due to differential expression. Thus, we conclude that substitutions within the RecA1 and RecA2 of Dhr1 that bypass *bud23Δ* do so by weakening its interaction with Utp14. Several mutated residues mapped to the interface between the two RecA domains while glutamate 1037, which is mutated to lysine or glutamine in two of the mutants, coordinates residues important for RNA binding. Although additional *in vitro* analysis is required to understand how these substitutions impact Dhr1 function, Dhr1-E1037K/Q likely impairs RNA binding.

Utp14 stimulates the unwinding activity of Dhr1 [30,32,65]. We previously reported five mutations within *UTP14* that suppress the growth defect of *bud23Δ* [30]. Notably, these residues are within an unresolved region of Utp14 spanning residues 719 to 780 that interacts with Dhr1 (S3A and S3B Fig), and extensive amino acid substitution or deletion of this region reduced Utp14 interaction with Dhr1, Utp14-dependent activation of Dhr1 unwinding activity *in vitro*, and phenocopied catalytically null Dhr1 *in vivo* [30]. Here, we report two additional mutations in *UTP14*, W791L and W794L, that suppressed *bud23Δ* (S3B Fig). These mutations are slightly downstream of those previously identified and affect two highly conserved tryptophan residues in a motif weakly reminiscent of a G patch, a motif common to activators of DEAH/RHA RNA helicases [30]. Determining how the mutations in *DHR1* and *UTP14* bypass loss of Bud23 will require further work. Nevertheless, the majority of the amino acid changes in Dhr1 and Utp14 are predicted to diminish their interaction and consequentially reduce the activity of Dhr1 (Fig 5) [30]. Thus, the simplest interpretation is that such mutants bypass *bud23Δ* by reducing Dhr1 activity, possibly increasing its opportunity to work on a suboptimal substrate (see Discussion).

## Depletion of Bud23 partially inhibits SSU Processome progression

The above genetic results suggest that disrupting protein-protein and protein-RNA interactions in the 3' basal subdomain of the SSU Processome can partially bypass the absence of Bud23. We interpret these results to indicate that Bud23 binding leads to disassembly events that promote the remodeling of this region the SSU Processome. To test the idea that Bud23 promotes disassembly events, we characterized pre-ribosomal particles in the absence of Bud23. We introduced a genomically encoded C-terminal Auxin-Induced Degron (AID) tag on Bud23 for rapid depletion of Bud23 upon the addition of the small molecule auxin without the need for shifting carbon sources [66]. The *BUD23-AID* strain grew similar to wild-type cells on media lacking auxin, while it showed a growth defect comparable to the *bud23Δ* strain on media containing auxin indicating that the AID tag is functional and does not obviously impact Bud23 function (Fig 6A). Bud23-AID was largely depleted after 10 minutes and undetectable after two hours of auxin treatment (Fig 6B).

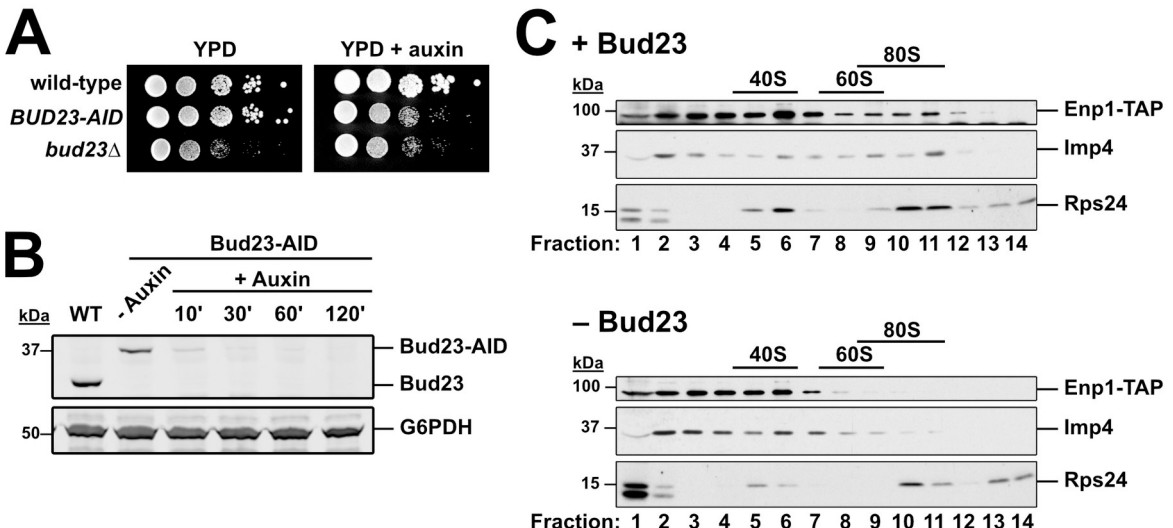

**Fig 6. Imp4 and Enp1 accumulate with pre-40S upon Bud23 depletion.** (A) The genomic fusion of an auxin-inducible degron (AID) to the C-terminus of Bud23 rendered cells sensitive to auxin, with a growth defect comparable to *bud23Δ*. 10-fold serial dilutions of wild-type (AJY2665), *BUD23-AID* (AJY4395), and *bud23Δ* (AJY3156) cells were spotted on YPD media with and without 0.5 mM auxin and grown for two days at 30°C. (B) Western blot of time-course of depletion of Bud23-AID, using equivalent amounts of total protein extract from AJY2665 or AJY4395 cells cultured to exponential phase then harvested prior to or after the addition of 0.5 mM auxin for the indicated time (WT; wild-type). G6PDH was used as a loading control. (C) The sucrose density gradient sedimentation of Enp1-TAP, Imp4, and Rps24 in the presence (upper panel) or absence (lower panel) of Bud23. Extracts were prepared from + Bud23 (AJY2665) and—Bud23 (AJY4395) cells treated with 0.5 mM auxin for two hours and separated on sucrose density gradients prior to fractionation. Proteins from each fraction were precipitated and subjected to Western blot analysis.

We first asked if Bud23 depletion impacted the association of Imp4 with pre-ribosomes. We depleted Bud23-AID for two hours and separated particles by ultracentrifugation in sucrose density gradients. We then fractionated the gradients, isolated the proteins from each fraction, and performed western blot analysis for Imp4. In the presence of Bud23, Imp4 sedimented throughout the gradient with enrichment in the 40S to 80S fractions and near the top of the gradient (Fig 6C). The Imp4 in fractions 2 and 3 likely reflects its association with the Mpp10 sub-complex [48,51,67] whereas the Imp4 in the 80S region reflects its association with the SSU Processome at 90S. We also monitored the sedimentation of the biogenesis factor Enp1 harboring a C-terminal Tandem Affinity Purification (TAP) tag which also sedimented throughout the gradient indicating its binding to both the SSU Processome and pre-40S particles [20]. In the absence of Bud23-AID, both Imp4 and Enp1-TAP showed reduced sedimentation in the 80S region but increased or maintained sedimentation in the 40S region of the gradient. The enrichment of these proteins in fractions 6 and 7 in the absence of Bud23 is reminiscent of the ~45S intermediate that accumulates in catalytically deficient Dhr1 mutants and are thought to represent a partially disassembled SSU Processome [29,30,46]. These results suggest that Imp4 and Enp1 are retained on a partially disassembled SSU Processome in the absence of Bud23, however we cannot exclude the possibility that the altered their sedimentation is due to degradation of unstable SSU Processome particles.

To further understand the nature of the 40S precursors that accumulate in the absence of Bud23, we affinity purified Enp1-TAP particles from WT and Bud23-AID strains after two hours of auxin treatment. Enp1-TAP is an ideal bait for these experiments as it binds to pre-ribosomes before Bud23 binding and is released after Bud23 function [20], and because its sucrose density gradient sedimentation was influenced by Bud23 (Fig 6C). Following enzymatic elution, the particles associated with Enp1 were sedimented through sucrose cushions to

separate them from any extraribosomal Enp1. The associated proteins were then separated by SDS-PAGE and analyzed by Coomassie staining. The depletion of Bud23 led to both the reduction of and accumulation of multiple protein species (Fig 7A; black lines and blue lines, respectively). Mass spectrometry of these discrete species identified the depleted proteins as the pre-40S factors Tsr1, Rio2, and Nob1, while mostly late-acting, SSU Processome factors comprised the accumulated factors suggesting that the loss of Bud23 precluded their release from pre-ribosomes.

To further characterize the purified particles, we used mass spectrometry to analyze the entire proteomic compositions of the wild-type and Bud23-depleted particles. To approximate stoichiometry for each 40S assembly protein, we calculated its relative spectral abundance factor (RSAF) by dividing the spectral counts for each protein by its molecular weight then normalizing this value to the bait, Enp1 (S8 Fig and S2 Table). This analysis showed various proteins that were accumulated or depleted from these particles. Interestingly, the proteins that showed genetic interaction with *bud23Δ* remained relatively constant in the absence of Bud23, with the exception of Utp2 which increased in abundance. As Enp1 and Imp4 co-sedimented in the absence of Bud23 (Fig 6C), we interpret this observation to indicate that these particles are arrested upstream of the release of these factors where they remain stoichiometric with Enp1. To simplify our analysis, we calculated a log2-fold change between Bud23-depleted and wild-type particles for each protein and considered a result significant if a protein showed a ± 0.5-fold change with a difference of at least 10 spectral counts (S9 Fig). This analysis revealed a set of 26 proteins that altered upon Bud23-depletion. These results agreed with what we observed by SDS-PAGE in Fig 7A in that there was an accumulation of mainly SSU Processome factors and a depletion of mostly pre-40S factors. Notably, Noc4, Utp2, and Rrp12 bind to the 3' major domain [15–17,19] while several factors, such as Bfr2, Enp2, Lcp5, Kre33, and Mrd1 have roles in the final assembly of the SSU Processome [19,68]. Strikingly, the pre-40S factors Slx9 and Rio1 [69–71] also accumulated in the absence of Bud23. These results suggest that without Bud23 the SSU Processome improperly disassembles in that certain regions are progressed to a pre-40S-like state whereas other regions appear arrested; however, whether this bottleneck represents a discrete SSU precursor or a heterogeneous mixture of precursors was not explored.

We also probed for the rRNA intermediates that co-precipitated with Enp1-TAP in the presence and absence of Bud23 (Fig 7B). We observed that Enp1 decreased association with the 23S rRNA with a concomitant accumulation of both 21S and 22S RNAs in the absence of Bud23. By quantifying the amount of 22S relative to 35S detected with A0-A1 probe vs the amount of 21S/22S relative to 35S detected with the A2-A3 probe we calculate that the 21S and 22S species are approximately equally abundant in the Enp1-TAP sample from Bud23-depleted cells. The accumulation of 21S and 22S indicates that processing at A2 was inhibited upon Bud23 depletion, as we have previously reported [39]. The accumulation of 22S was unexpected as this indicates a defect in A1 cleavage as well. Furthermore, we saw a modest accumulation of the A0-cleaved 5' ETS rRNA (~1.5-fold) and U3 snoRNA (~1.2-fold). Consistent with what is seen in whole cell extracts of *bud23Δ* cells (Figs 2E and 4B) [45,46], we also saw that 27SA2 intermediate was present in the input and IP for the wild-type sample, but totally absent in the Bud23-depleted sample (Fig 7B). These results reconfirm that Bud23 is needed for efficient rRNA processing.

## Discussion

Bud23 is the methyltransferase that modifies G1575 in 18S rRNA and is thought to act at a relatively late stage of nucleolar SSU assembly [39,42]. Although it is conserved from yeast to

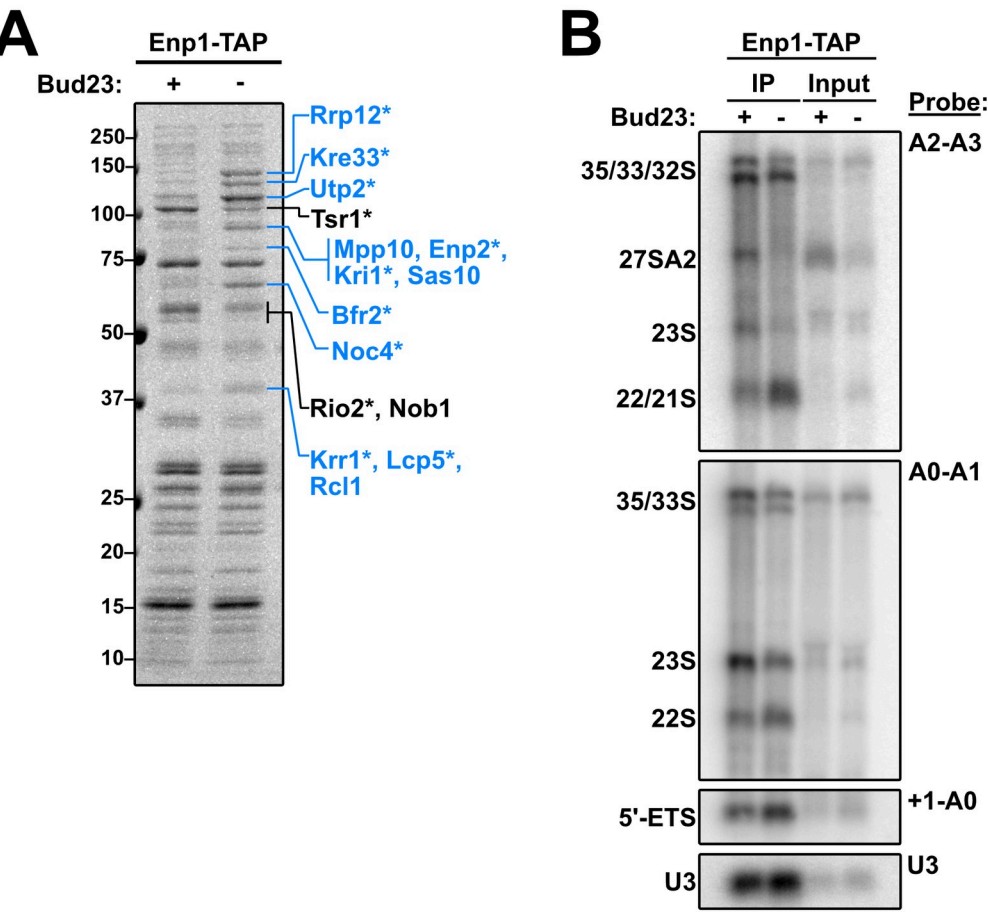

**Fig 7. Composition of 40S precursors purified in the absence of Bud23.** (A) Coomassie-stained gel of proteins that co-purified with Enp1-TAP in the presence (+) or absence (-) of Bud23. Pre-ribosomal particles were enriched by overlaying eluate onto sucrose cushions followed by ultracentrifugation. Individual species that showed clear enrichment or depletion were excised and identified by mass spectrometry and are indicated in blue or black text, respectively. The asterisks (*) denote proteins that also appeared in the analysis described in S9 Fig. (B) The rRNA processing intermediates and U3 snoRNA that co-purified with Enp1-TAP in the presence (+) or absence (-) of Bud23 were detected by Northern blotting using the indicated probes. Oligonucleotides are listed in Table 3.

humans [40,72–74] and deletion of *BUD23* in yeast leads to severely impaired growth [39], its methyltransferase activity is dispensable for ribosome assembly [39,72]. This indicates that the primary function of Bud23 stems from its binding to SSU precursors, however this role is not well understood. Bud23 and its cofactor Trm112 co-sediment with both the pre-40S and the SSU Processome [40,42,45,46]. Consistent with this, Bud23 also co-purifies with late-acting SSU Processome factors and early pre-40S factors [45,46]. Although Bud23 is not a stable component of the SSU Processome and, in fact, its binding site is occupied by the assembly factor Emg1 in the SSU Processome structures [15–17], our work implicates Bud23 in the disassembly of the SSU Processome as it transitions to a pre-40S. The absence of Bud23 impairs A2 site cleavage [39], an event that is tied to the transition. Prior to A2 cleavage, the U3 snoRNA is unwound from the rRNA which is catalyzed by the RNA helicase Dhr1 and its co-factor Utp14 [29,30]. Dhr1 physically interacts with Bud23 [40,46], and we previously reported that mutations in *DHR1*, *UTP14*, or other SSU Processome factors suppress the growth and A2 site cleavage defects in *bud23Δ* cells [30,45–47]. These results indicate that Bud23 enters the SSU assembly pathway as the SSU Processome is transitioning to the pre-40S.

## Does Bud23 promote the final step of the SSU Processome disassembly?

The 3' basal subdomain forms a compact RNA unit whose structure remains relatively unchanged between the SSU Processome and Bud23-bound pre-40S, despite its dramatic repositioning during this transition. The majority of mutations that we found as suppressors of *bud23Δ* were in *IMP4* and *DHR1* with additional mutations in *BMS1*, *RPS28A*, *UTP2*, and *UTP14* (Fig 1D and 1E). Imp4, Rps28, Utp2 and Bms1 all interact with the 3' basal subdomain of nascent 18S rRNA. The mutated residues in Imp4 clustered primarily in its interface with the RNA of the 3' basal subdomain (Fig 2C), opposite the binding site of Bud23 and are predicted to be disrupting interactions. The Brix domain of Imp4 also contacts the NTD of Utp2 (Fig 3C), and we showed that suppressing amino acid changes on either side of this interface disrupted their interaction (Fig 3D and 3F).

While this manuscript was in review, the structures of several SSU Processome disassembly intermediates were published [75,76]. These structures reveal a stepwise disassembly that starts with the cleavage of the A1 site and continues with the successive shedding of most SSU Processome factors and the concomitant partial compaction of the rRNA domains. This process culminates in a partially disassembled SSU Processome complex, termed the "Dis-C" complex, in which the majority of SSU Processome factors have been released. Only 15 SSU Processome factors are resolved in the Dis-C structure [76]. Strikingly, the U3 snoRNA and all six of the proteins in which we identified suppressors of *bud23Δ* are retained on the Dis-C complex (Fig 8A), and we suggest that this complex is close to what Bud23 binds. In the Dis-C structure the 3' major domain, encompassing the 3' basal subdomain, is partially rotated relative to its position in the complete SSU Processome (Fig 8B), but has not yet fully rotated to adopt its nearly mature position observed in the Bud23-bound early pre-40S particle [44]. Interestingly, this rotation is sterically incompatible with the presence of Imp4 and Utp2 suggesting that these two proteins must be released for this final rotation of the 3' major domain needed to produce a pre-40S. Notably, the Utp2-Imp4 interaction (Fig 3C–3F) and the helical extensions of Bms1 and Imp4 that embrace the U3-rRNA duplexes in the complete SSU Processome (Fig 1D) are not seen in the Dis-C complex [76], suggesting that it is poised to complete the transition to pre-40S. The picture that emerges is that the *bud23Δ* suppressing mutations in *IMP4*, *BMS1*, *UTP2* and *RPS28A* define a network of protein-protein and protein-RNA contacts that stabilize the 3' basal subdomain within the transitioning structure. Disrupting this interaction network bypasses the requirement for Bud23, suggesting that the function of Bud23 is to destabilize these interactions to promote progression to the pre-40S. Consistent with this idea, in the absence of Bud23 pre-ribosomal particles were enriched for late-acting SSU Processome factors and depleted of pre-40S factors (Fig 7A and S9 Fig). Thus, we propose that Bud23 binding to the 3' basal subdomain induces the rotation of the 3' major domain as the Dis-C complex transitions into the pre-40S and consequentially promotes the final disassembly of the SSU Processome (Fig 8).

## Are the enzymatic activities of Bms1 and Dhr1 coordinated?

The network of genetic and physical interactions that we define in this work functionally link the GTPase Bms1 and the RNA helicase Dhr1 whose enzymatic activities are thought to be critical for the transition of the SSU Processome to a pre-40S particle. While the exact molecular function of Bms1 remains unknown, its GTPase activity is essential [57,58,60,61]. Dhr1 is proposed to catalyze U3 snoRNA removal by binding just downstream of the U3-rRNA duplexes [29] where it is poised to translocate in a 3' to 5' manner [32]. Indeed, the Dis-C structure confirms this notion [76]. The catalytic domains of Bms1 and Dhr1 are positioned on opposite sides of the Dis-C structure; however, the N-terminus of Dhr1 extends to interact with Bms1 at the distal end of the U3-rRNA duplex that Dhr1 unwinds (S10 Fig). In this way, Bms1 and Dhr1 can be physically and

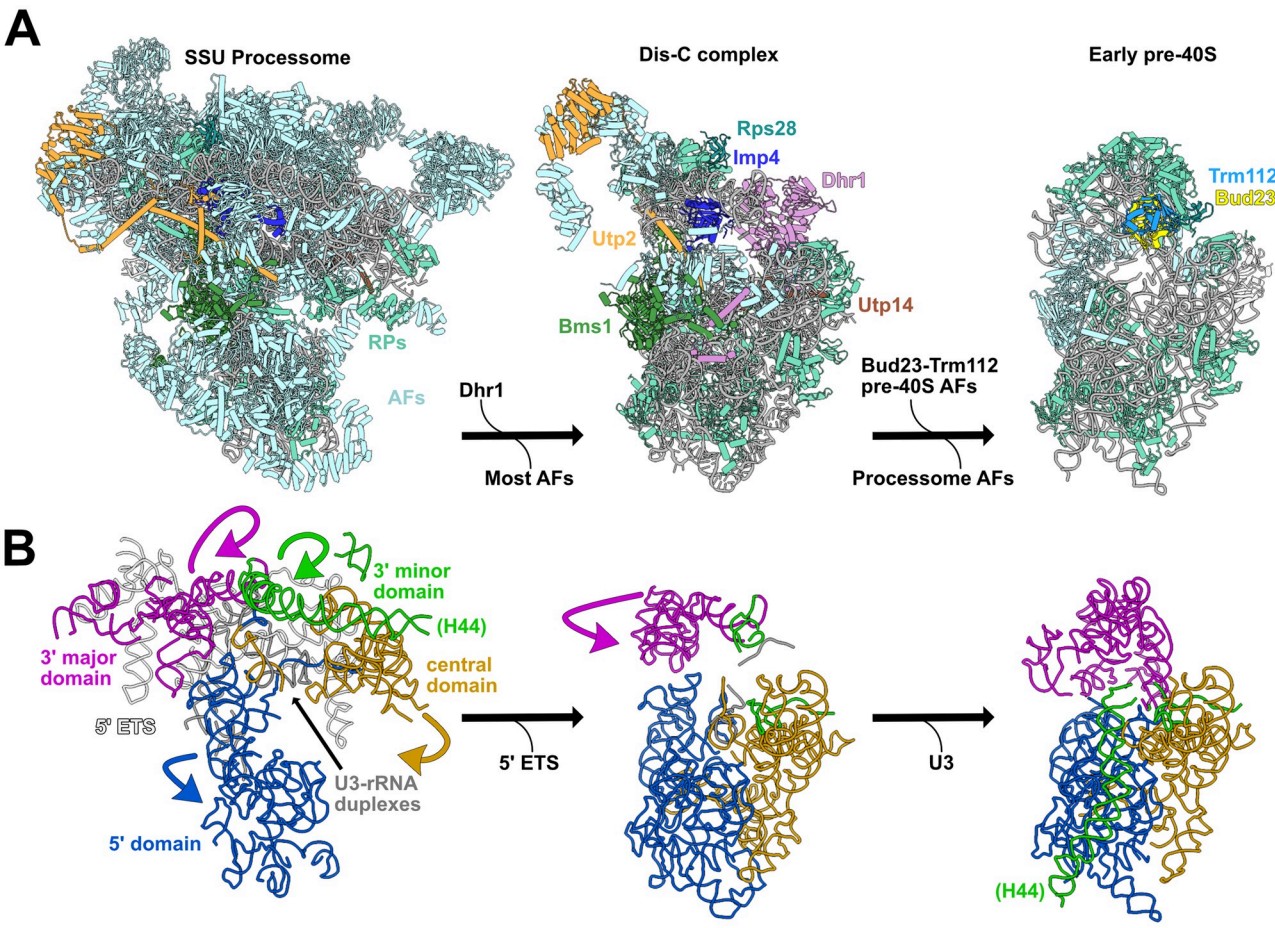

**Fig 8. Model for when Bud23 functions during SSU Processome progression.** (A) The complete SSU Processome is decorated with assembly factors (AFs; light blue) and some ribosomal proteins (RPs; teal) that establish the rRNA architecture. During the transition to the Dis-C complex Dhr1 is recruited and most SSU Processome AFs are released. The Dis-C complex harbors 15 AFs including Rps28 (cyan), Imp4 (blue), Utp2 (orange), Bms1 (green), Utp14 (brown) and Dhr1 (pink). The incorporation of Bud23-Trm112 (light yellow and light blue, respectively) promotes the release of the residual SSU Processome AFs, resulting in the final disassembly of the SSU Processome to generate an early pre-40S complex. (B) In the complete SSU Processome, the rRNA of the 5' (blue), central (yellow), 3' major (magenta), and 3' minor (green) domains are splayed apart by the U3 snoRNA (gray) and the 5' ETS (white). These domains become partially compacted and the 5' ETS is released during the transition to the Dis-C complex. The release of the U3 snoRNA and the further rotation of the 3' major and 3' minor domains produces the early pre-40S. The yeast SSU Processome (left) is a composite PDBs 5WLC and 5WYK, the Dis-C complex (middle) is from PDB 6ZQG, and the early pre-40S (right) is from humans (PDB 6G4W). Images were generated in UCSF ChimeraX v0.93 [88].

mechanistically linked. Notably, Bms1 remains in its GTP-bound state in the Dis-C complex [76]. As the Bms1 substitutions that suppress *bud23Δ* are expected to promote a conformational change within Bms1 (Fig 4F), we speculate that Bud23 binding signals to activate Bms1. The linkage of Dhr1 and Bms1 in the disassembly of the SSU Processome is reminiscent of the relationship between the DExH RNA helicase Brr2 and the EF-G-like GTPase Snu114 that regulate spliceosome disassembly [77]. In the case of the spliceosome, it is believed that nucleotide-dependent conformational changes in Snu114 activate Brr2. It is possible that the actions of Dhr1 and Bms1 are similarly coordinated in that Bms1 could relay a signal, such as Bud23 binding, to Dhr1.

## How do the Dhr1 and Utp14 mutants bypass *bud23Δ*?

The activity of Dhr1 is stimulated by interaction with Utp14 both *in vivo* and *in vitro* [30,32,65] where Utp14 is thought to act as a processivity factor [32]. This interaction is mediated by a

short loop within Utp14, spanning residues 719–780 in the yeast protein [30], that is conserved from yeast to mammals [30,32,65]. Y2H and *in vitro* experiments show that substantial mutation of this loop disrupts the Utp14-Dhr1 interaction and decreases stimulation of Dhr1 activity [30,32]. Consistent with the *in vitro* results, these Utp14 mutants phenocopy a catalytically defective Dhr1 mutant *in vivo* [30]. Single amino acid substitutions within this loop were identified as suppressors of *bud23Δ*, suggesting that weakening the interaction between Utp14 and Dhr1 and likewise reducing the activation of Dhr1 by Utp14 suppresses *bud23Δ*. Similarly, the majority of the substitutions in Dhr1 that suppress *bud23Δ* lay across the surface of the RecA domains (Fig 5C) and combining multiple substitutions leads to reduced interaction with Utp14 (Fig 5D). The ATPase activity of Dhr1 also depends on its interaction with RNA [29,30,65], and other *bud23Δ*-suppressing mutations, such as *dhr1-E1037K/Q*, affect residues that appear important for RNA binding. As with the Utp14 mutants, it appears that reducing Dhr1 activity bypasses *bud23Δ*.

Formation of the central pseudoknot (CPK) requires the release of the U3 snoRNA from the rRNA as well as the hybridization of nucleotides 1137–1144 of the 18S rRNA with its 5' end to form helix 2. The rotation of the 3' major domain appears crucial for bringing the strands of helix 2 together. If Bud23 promotes the structural rearrangements necessary to position the RNAs for CPK folding, then Dhr1 unwinding of U3 may be largely unproductive in the absence of Bud23; acting before substrates are correctly positioned and failing in the progression towards the pre-40S. Thus, amino acid changes in Dhr1 or Utp14 that reduce Dhr1 activity, may afford more time for rearrangements within the transitioning SSU Processome to occur and allow for productive U3 unwinding by Dhr1. Similar models of kinetic proofreading are well-established for RNA helicases involved in splicesosomal rearrangements [78] where, for example, mutations in Prp28 that reduce its ATPase activity enhance the splicing efficiency of a suboptimal substrate [79].

## Materials and methods

### Strains, growth media, genetic methods, and yeast two-hybrid (Y2H) analysis

All *S. cerevisiae* strains and sources are listed in Table 1. AJY2676 was generated by genomic integration of *Eco*RI-digested pAJ4339 [80] into AJY2161 to replace the *KanMX* marker with *CloNAT*. AJY3156 was generated by genomic integration of *bud23Δ::KanMX* into AJY2665. AJY3822 was generated by genomic integration of *KanMX::PGAL1-3xHA* into the diploid strain BY4743 (Open Biosystems), sporulated, and dissected. AJY4175 was generated by transforming pAJ4094 into a *UTP2/utp2Δ::KanMX* heterozygous diploid strain [81], sporulated, and dissected. AJY4395 was generated by genomic integration of *AID-HA::OsTIR1::LEU2* amplified from pJW1662 [82] into the *BUD23* locus of AJY2665. AJY4605 was generated by crossing AJY3244 (*MATalpha KanMX::PGAL1-3xHA-UTP14*; this study) with AJY4387 (*MATa CloNAT::PGAL1-3xHA-DHR1*; this study), sporulated, and dissected. All yeast strains were cultured at 30°C in either YPD (2% peptone, 1% yeast extract, 2% dextrose), YPgal (2% peptone, 1% yeast extract, 1% galactose), or synthetic dropout (SD) medium containing 2% dextrose unless otherwise noted. When appropriate, media were supplemented with 150 to 250 μg/ml G418 or 100 μg/ml nourseothricin. All plasmids and sources are listed in Table 2. Y2H analysis was performed as previously described [33].

### Identification of additional spontaneous suppressors of *bud23Δ*

AJY2676 cells were inoculated into 200 μL of YPD media in a 48-well format plate. Cells were cultured with continuous shaking until saturation then diluted into fresh media. After each

**Table 1. Yeast strains used in this study.**

| Strain | Genotype | Reference |
|--------|----------|-----------|
| AJY2161 | *MATa his3Δ1 leu2Δ0 ura3Δ0 lys2Δ0 met15Δ0 bud23Δ::KanMX* | [39] |
| AJY2665 | *MATa his3Δ1 leu2Δ0 met15Δ0 ura3Δ0 ENP1-TAP::HIS3MX6* | [89] |
| AJY2676 | *MATa his3Δ1 leu2Δ0 ura3Δ0 lys2Δ0 met15Δ0 bud23Δ::CloNAT* | This study & [47]. |
| AJY3156 | *MATa his3Δ1 leu2Δ0 met15Δ0 ura3Δ0 ENP1-TAP::HIS3MX6 bud23Δ::KanMX* | This study. |
| AJY3512 | *MATa his3Δ1 leu2Δ0 ura3Δ0 lys2Δ0 met15Δ0 bud23Δ::KanMX imp4-V170F* | This study & [47]. |
| AJY3579 | *MATa his3Δ1 leu2Δ0 ura3Δ0 lys2Δ0 met15Δ0 bud23Δ::KanMX imp4-R94L* | This study & [47]. |
| AJY3580 | *MATa his3Δ1 leu2Δ0 ura3Δ0 lys2Δ0 met15Δ0 bud23Δ::KanMX imp4-N118K* | This study & [47]. |
| AJY3581 | *MATa his3Δ1 leu2Δ0 ura3Δ0 lys2Δ0 met15Δ0 bud23Δ::KanMX utp2-A2D* | [45] |
| AJY3741 | *MATa his3Δ1 leu2Δ0 ura3Δ0 lys2Δ0 met15Δ0 bud23Δ::KanMX imp4-T92I* | This study & [47]. |
| AJY3742 | *MATa his3Δ1 leu2Δ0 ura3Δ0 lys2Δ0 met15Δ0 bud23Δ::KanMX imp4-R116M* | This study & [47]. |
| AJY3743 | *MATa his3Δ1 leu2Δ0 ura3Δ0 lys2Δ0 met15Δ0 bud23Δ::KanMX imp4-S93T* | This study & [47]. |
| AJY3744 | *MATa his3Δ1 leu2Δ0 ura3Δ0 lys2Δ0 met15Δ0 bud23Δ::KanMX imp4-R94S* | This study & [47]. |
| AJY3745 | *MATa his3Δ1 leu2Δ0 ura3Δ0 lys2Δ0 met15Δ0 bud23Δ::KanMX imp4-N118D* | This study & [47]. |
| AJY3822 | *MATa his3Δ1 leu2Δ0 ura3Δ0 met15Δ0 KanMX::PGAL1-3xHA-IMP4* | This study. |
| AJY4175 | *MATa his3Δ1 leu2Δ0 ura3Δ0 utp2Δ::KanMX (PGAL10-UTP2 LEU2 CEN ARS)* | This study. |
| AJY4377 | *MATa his3-1 leu2-0 met15-0 pBMS1::kanR-tet07-TATA URA3::CMV-tTA* | [90] |
| AJY4395 | *MATa his3Δ1 leu2Δ0 met15Δ0ura3Δ0 ENP1-TAP::HIS3MX6 BUD23-AID-HA:: OsTIR1::LEU2* | This study. |
| AJY4501 | *MATa his3Δ1 leu2Δ0 ura3Δ0 lys2Δ0 met15Δ0 bud23Δ::CloNAT imp4-H159R* | This study. |
| AJY4502 | *MATa his3Δ1 leu2Δ0 ura3Δ0 lys2Δ0 met15Δ0 bud23Δ::CloNAT utp2-L9S* | This study. |
| AJY4503 | *MATa his3Δ1 leu2Δ0 ura3Δ0 lys2Δ0 met15Δ0 bud23Δ::CloNAT utp14-W794L* | This study. |
| AJY4504 | *MATa his3Δ1 leu2Δ0 ura3Δ0 lys2Δ0 met15Δ0 bud23Δ::CloNAT imp4-R99L* | This study. |
| AJY4505 | *MATa his3Δ1 leu2Δ0 ura3Δ0 lys2Δ0 met15Δ0 bud23Δ::CloNAT imp4-N121I* | This study. |
| AJY4506 | *MATa his3Δ1 leu2Δ0 ura3Δ0 lys2Δ0 met15Δ0 bud23Δ::CloNAT imp4-Y77C* | This study. |
| AJY4507 | *MATa his3Δ1 leu2Δ0 ura3Δ0 lys2Δ0 met15Δ0 bud23Δ::CloNAT imp4-S101W* | This study. |
| AJY4508 | *MATa his3Δ1 leu2Δ0 ura3Δ0 lys2Δ0 met15Δ0 bud23Δ::CloNAT imp4-H208D* | This study. |
| AJY4509 | *MATa his3Δ1 leu2Δ0 ura3Δ0 lys2Δ0 met15Δ0 bud23Δ::CloNAT imp4-R94C* | This study. |
| AJY4510 | *MATa his3Δ1 leu2Δ0 ura3Δ0 lys2Δ0 met15Δ0 bud23Δ::CloNAT imp4-H156D* | This study. |
| AJY4511 | *MATa his3Δ1 leu2Δ0 ura3Δ0 lys2Δ0 met15Δ0 bud23Δ::CloNAT imp4-R99H* | This study. |
| AJY4512 | *MATa his3Δ1 leu2Δ0 ura3Δ0 lys2Δ0 met15Δ0 bud23Δ::CloNAT imp4-R146G* | This study. |
| AJY4513 | *MATa his3Δ1 leu2Δ0 ura3Δ0 lys2Δ0 met15Δ0 bud23Δ::CloNAT W791L* | This study. |
| AJY4514 | *MATa his3Δ1 leu2Δ0 ura3Δ0 lys2Δ0 met15Δ0 bud23Δ::CloNAT dhr1-D408Y* | This study. |
| AJY4515 | *MATa his3Δ1 leu2Δ0 ura3Δ0 lys2Δ0 met15Δ0 bud23Δ::CloNAT dhr1-G432R* | This study. |
| AJY4516 | *MATa his3Δ1 leu2Δ0 ura3Δ0 lys2Δ0 met15Δ0 bud23Δ::CloNAT dhr1-S511Y* | This study. |
| AJY4517 | *MATa his3Δ1 leu2Δ0 ura3Δ0 lys2Δ0 met15Δ0 bud23Δ::CloNAT dhr1-G434D* | This study. |
| AJY4518 | *MATa his3Δ1 leu2Δ0 ura3Δ0 lys2Δ0 met15Δ0 bud23Δ::CloNAT dhr1-E397D* | This study. |
| AJY4519 | *MATa his3Δ1 leu2Δ0 ura3Δ0 lys2Δ0 met15Δ0 bud23Δ::CloNAT dhr1-R596G* | This study. |
| AJY4520 | *MATa his3Δ1 leu2Δ0 ura3Δ0 lys2Δ0 met15Δ0 bud23Δ::CloNAT dhr1-D566Y* | This study. |
| AJY4521 | *MATa his3Δ1 leu2Δ0 ura3Δ0 lys2Δ0 met15Δ0 bud23Δ::CloNAT dhr1-A804D* | This study. |
| AJY4522 | *MATa his3Δ1 leu2Δ0 ura3Δ0 lys2Δ0 met15Δ0 bud23Δ::CloNAT dhr1-R596S* | This study. |
| AJY4523 | *MATa his3Δ1 leu2Δ0 ura3Δ0 lys2Δ0 met15Δ0 bud23Δ::CloNAT dhr1-M857I* | This study. |

(*Continued*)

**Table 1.** (Continued)

| Strain | Genotype | Reference |
|---|---|---|
| AJY4524 | *MATa his3Δ1 leu2Δ0 ura3Δ0 lys2Δ0 met15Δ0 bud23Δ::CloNAT dhr1-R563M* | This study. |
| AJY4525 | *MATa his3Δ1 leu2Δ0 ura3Δ0 lys2Δ0 met15Δ0 bud23Δ::CloNAT dhr1-R13G* | This study. |
| AJY4526 | *MATa his3Δ1 leu2Δ0 ura3Δ0 lys2Δ0 met15Δ0 bud23Δ::CloNAT dhr1-E1037Q* | This study. |
| AJY4527 | *MATa his3Δ1 leu2Δ0 ura3Δ0 lys2Δ0 met15Δ0 bud23Δ::CloNAT dhr1-E1037K* | This study. |
| AJY4529 | *MATa his3Δ1 leu2Δ0 ura3Δ0 lys2Δ0 met15Δ0 bud23Δ::CloNAT bms1-D843V* | This study. |
| AJY4530 | *MATa his3Δ1 leu2Δ0 ura3Δ0 lys2Δ0 met15Δ0 bud23Δ::CloNAT dhr1-F594L* | This study. |
| AJY4531 | *MATa his3Δ1 leu2Δ0 ura3Δ0 lys2Δ0 met15Δ0 bud23Δ::CloNAT rps28a-G24D* | This study. |
| AJY4532 | *MATa his3Δ1 leu2Δ0 ura3Δ0 lys2Δ0 met15Δ0 bud23Δ::CloNAT bms1-D124Y* | This study. |
| AJY4533 | *MATa his3Δ1 leu2Δ0 ura3Δ0 lys2Δ0 met15Δ0 bud23Δ::CloNAT bms1-A903P* | This study. |
| AJY4535 | *MATa his3Δ1 leu2Δ0 ura3Δ0 lys2Δ0 met15Δ0 bud23Δ::CloNAT bms1-G813S* | This study. |
| AJY4536 | *MATa his3Δ1 leu2Δ0 ura3Δ0 lys2Δ0 met15Δ0 bud23Δ::CloNAT imp4-P252L* | This study. |
| AJY4537 | *MATa his3Δ1 leu2Δ0 ura3Δ0 lys2Δ0 met15Δ0 bud23Δ::CloNAT bms1-S1020L* | This study. |
| AJY4605 | *MATa his3Δ1 leu2Δ0 ura3Δ0 met15Δ0 KanMX::PGAL1-3xHA-UTP14 CloNAT:: PGAL1-3xHA-DHR1* | This study. |
| BY4741 | *MATa his3Δ1 leu2Δ0 met15Δ0 ura3Δ0* | Open Biosystems |
| PJ69-4a | *MATa trp1-901 leu2-3,112 ura3-52 his3-200 gal4Δ gal80Δ LYS2::GAL1-HIS3 GAL2-ADE2 met2::GAL7-lacZ* | [91] |
| PJ69-4alpha | *MATalpha trp1-901 leu2-3,112 ura3-52 his3-200 gal4Δ gal80Δ LYS2::GAL1-HIS3 GAL2-ADE2 met2::GAL7-lacZ* | [91] |

passage, cells were plated onto YPD plates to test for the presence of suppressors. This process was iterated for each culture until suppressors were observed. Single colonies of each suppressor strain were obtained, and genomic DNA was prepped using MasterPure Yeast DNA Purification Kit (Lucigen). The *DHR1*, *IMP4*, *UTP2*, and *UTP14* loci were amplified and sequenced by Sanger to identify mutations in known suppressors [30,45–47].

Libraries for the six strains that did not carry suppressors in *DHR1*, *IMP4*, *UTP2* or *UTP14* were prepared and sequenced on an Illumina NextSeq 500 platform by the Genomic Sequencing and Analysis Facility at the University of Texas at Austin. The quality of the resultant reads was assessed using FastQC (v0.10.1) [http://www.bioinformatics.babraham.ac.uk/projects/fastqc/], and subsequently processed using TrimGalore (v1.14) [http://www.bioinformatics.babraham.ac.uk/projects/trim_galore/] to discard low-quality sequences and adapters. The processed reads were aligned using Bowtie2 (v2.3.4) [83] using the default settings for paired-end reads. The resultant files were further processed with SAMtools (v0.1.18) [84] and BCFtools (v0.1.17) [85] to generate variant call format files. VCFtools (v0.1.16) [86] was used to filter out variants with low quality scores (Quality value < 100) and to compare samples pairwise to identify mutations unique to each suppressed strain. This analysis revealed single point mutations within *BMS1* and *RPS28A* that were confirmed by Sanger sequencing. The *bms1* and *rps28A* variants were subsequently cloned into centromeric vectors and, as with all other *bud23Δ* suppressors that we tested, the *rps28A* and *bms1* mutants were dominant. All mutant strains isolated in this screen are listed in Table 1.

## Identification of mutations in *UTP2* that suppress *bud23Δ*

Random mutations in *UTP2* were generated by error-prone PCR using *Taq* polymerase and pAJ2595 as the template and oligos that hybridize to the upstream and downstream sequences of *UTP2*. The restriction enzymes *Eco*RI and *Sph*I were used to linearize the vector pAJ2595

**Table 2. Plasmids used in this study.**

| Plasmid | Description | Reference |
|---|---|---|
| pAJ2321 | GAL4AD-HA-UTP14 LEU2 2µ | [30] |
| pAJ2595 | UTP2 URA3 CEN ARS | [45] |
| pAJ2596 | utp2-A2D URA3 CEN ARS | [45] |
| pAJ2720 | IMP4-13xMYC LEU2 CEN ARS | This study. |
| pAJ2723 | imp4-S93T, R94S, R116M, N118D-13xMYC LEU2 CEN ARS | This study. |
| pAJ2910 | IMP4 LEU2 CEN ARS | This study & [47]. |
| pAJ2911 | imp4-S93T, R94S LEU2 CEN ARS | This study & [47]. |
| pAJ2912 | imp4-R116M, N118D LEU2 CEN ARS | This study & [47]. |
| pAJ2922 | GAL4BD-c-myc-DHR1 TRP1 2µ | [46] |
| pAJ2923 | imp4-S93T, R94S, R116M, N118D LEU2 CEN ARS | This study & [47]. |
| pAJ2927 | imp4-R94S LEU2 CEN ARS | This S study & [47]. |
| pAJ2928 | imp4-R116M LEU2 CEN ARS | This study & [47]. |
| pAJ2929 | imp4-N118D LEU2 CEN ARS | This study & [47]. |
| pAJ2769 | GAL4BD-c-myc-IMP4 TRP1 2µ | This study. |
| pAJ3082 | DHR1 LEU2 CEN ARS | [46] |
| pAJ3332 | utp2-F149S URA3 CEN ARS | This S study. |
| pAJ3335 | utp2-L151H URA3 CEN ARS | This study. |
| pAJ3347 | utp2-L148S, F149S, L151H URA3 CEN ARS | This study. |
| pAJ3348 | utp2-L6P URA3 CEN ARS | This study. |
| pAJ3349 | utp2-K7E URA3 CEN ARS | This study. |
| pAJ3350 | utp2-L148S URA3 CEN ARS | This study. |
| pAJ4093 | utp2-F58S URA3 CEN ARS | This study. |
| pAJ4094 | PGAL10-UTP2 LEU2 CEN ARS | This study. |
| pAJ4095 | utp2-A2D, L6P, K7E URA3 CEN ARS | This study. |
| pAJ4153 | dhr1-R563M LEU2 CEN ARS | This study. |
| pAJ4188 | Utp2-GAL4AD-HA LEU2 2µ | This study. |
| pAJ4192 | utp2-F58S-GAL4AD-HA LEU2 2µ | This study. |
| pAJ4193 | utp2-L148S, F149S, L151H-GAL4AD-HA LEU2 2µ | This study. |
| pAJ4194 | utp2-A2D, L6P, K7E-GAL4AD-HA LEU2 2µ | This study. |
| pAJ4493 | GAL4BD-c-myc-imp4-V170F TRP1 2µ | This study. |
| pAJ4494 | GAL4BD-c-myc-imp4-P252L TRP1 2µ | This study. |
| pAJ4475 | BMS1 LEU2 CEN ARS | This study. |
| pAJ4476 | bms1-D124Y LEU2 CEN ARS | This study. |
| pAJ4477 | bms1-G813S LEU2 CEN ARS | This study. |
| pAJ4503 | GAL4BD-c-myc-DHR1-E360K, E397D, E402G, D408Y TRP1 2µ | This study. |
| pAJ4513 | GAL4BD-c-myc-DHR1-R563M, D566Y, E831K, F837L TRP1 2µ | This study. |
| pAJ4514 | GAL4BD-c-myc-DHR1-H593Y, R596C, E831K TRP1 2µ | This study. |
| pAJ4664 | dhr1-E360K LEU2 CEN ARS | This study. |
| pAJ4665 | dhr1-E402G LEU2 CEN ARS | This study. |
| pAJ4666 | dhr1-E430K LEU2 CEN ARS | This study. |
| pAJ4667 | dhr1-D566Y LEU2 CEN ARS | This study. |
| pAJ4668 | dhr1-M744T LEU2 CEN ARS | This study. |
| pAJ4669 | dhr1-A804D LEU2 CEN ARS | This study. |
| pAJ4670 | dhr1-F837L LEU2 CEN ARS | This study. |
| pAJ4671 | dhr1-E1037K LEU2 CEN ARS | This study. |
| pGADT7 | GAL4AD-HA LEU2 2µ | [92] |
| pGBKT7 | GAL4BD-c-myc TRP1 2µ | [92] |

(*Continued*)

**Table 2.** (Continued)

| Plasmid | Description | Reference |
|---|---|---|
| pJW1662 | *AtIAA17_71-114(AID*)-HA::LEU2 prADH1-OsTIR1* | [82] |
| pRS315 | *LEU2 CEN ARS* | [93] |
| pRS415 | *LEU2 CEN ARS* | [93] |
| pRS416 | *URA3 CEN ARS* | [93] |

and the linearized vector was, co-transformed with the mutant amplicon into AJY2161, and plated onto SD-Uracil media to allow recombination of the mutant amplicon into the pAJ2595 backbone. Colonies displaying a suppressed phenotype were isolated; vectors were rescued from yeast and sequenced after confirming that the vectors conferred suppression.

## Affinity purification

Cells were cultured as described in the Northern blotting and mass spectrometry subsections below. All steps were carried out on ice or at 4°C. Cells were washed with Lysis Buffer (50 mM Tris-HCl pH 7.6 (25°C), 100 mM KCl, 5 mM $MgCl_2$, 5 mM beta-mercaptoethanol (βME), 1 mM each of PMSF and Benzamidine, and 1 μM each of leupeptin and pepstatin) supplemented with EDTA-free Pierce Protease Inhibitor Mini Tablet cocktail (Thermo Scientific), then resuspended in 1 volume Lysis Buffer. Extracts were generated by glass bead lysis and clarified at 18,000$g$ for 15 minutes. Clarified extracts were normalized according to $A_{260}$ and supplemented with 0.1% TritonX-100. Normalized extracts were incubated for 1.5 hours with rabbit IgG (Sigma) coupled to Dynabeads (Invitrogen), prepared as previously described [87]. Following binding, the beads were washed thrice with Wash Buffer (Lysis Buffer supplemented with 0.1% TritonX-100). The beads were resuspended in Elution Buffer (Wash Buffer supplemented with TEV protease and Murine RNase Inhibitor (New England Biolabs)) and the bound Enp1-TAP containing complexes were eluted for 1.5–2 hours. The resultant eluates were handled as described in the Northern blotting and mass spectrometry subsections below.

## Northern blot analysis

For analysis of rRNA processing in whole cell extract (WCE), strains were cultured overnight in YPD media to saturation. Cell cultures were diluted into YPD at a starting $OD_{600}$ of 0.1 and cultured to mid-exponential phase ($OD_{600}$ ~0.4–0.5) before collection and storage at -80°C prior to lysis. For analysis of affinity purified RNAs, strains AJY2665 and AJY4395 were cultured overnight in YPD media to saturation. Cells were diluted into YPD at a starting $OD_{600}$ of 0.05 and cultured for three hours. Cultures were treated with 0.5 mM auxin for 2 hours at 30°C, centrifuged, and frozen in liquid nitrogen. Affinity purification was performed as described above. Affinity purified and WCE RNAs were isolated using the acid-phenol-chloroform method as previously described (Zhu et al. 2016). RNAs were electrophoresed through 1.2%-agarose MOPS 6% formaldehyde gel. Northern blotting was performed as previously described (Li et al. 2009) using the oligo probes listed in Table 3, and signal was detected by phosphoimaging on a GE Typhoon FLA9500.

## Mass spectrometry and analysis

Strains AJY2665 and AJY4395 were cultured as described in the Northern blot analysis subsection. Affinity purifications were performed as described above. To isolate factors associated with only pre-ribosomal particles, the eluate was overlaid onto a sucrose cushion (15% D-

**Table 3. Oligonucleotide probes used for Northern blotting.**

| Target | Sequence |
|---|---|
| +1-A0 | 5' GGTCTCTCTGCTGCCGGAAATG 3' |
| A0-A1 | 5' CCCACCTATTCCCTCTTGC 3' |
| A2-A3 | 5' TGTTACCTCTGGGCCCCGATTG 3' |
| U3 snoRNA | 5' TAGATTCAATTTCGGTTTCTC 3' |

sucrose, 50 mM Tris-HCl pH 7.6 (25°C), 100 mM KCl, 5 MgCl$_2$) then centrifuged at 70,000 rpm for 15 min in a Beckman Coulter TLA100 rotor. Following, the pellets were precipitated with 15% trichloroacetic acid (TCA), washed with acetone and dried, and resuspended in 1X Laemmli buffer. Approximately equivalent amounts of protein were either fully separated on SDS-PAGE gels for excision of individual species or electrophoresed slightly into a NuPAGE Novex 4%–12% Bis-Tris gel for analysis of the entire affinity purification. Peptides were recovered from in-gel Trypsin digestion and prepared for identification by mass spectrometry as previously described [33]. The resultant peptides were identified at The University of Texas at Austin Proteomics Facility by LC-MS/MS on a Thermo Orbitrap Fusion 1 with either a 30 minute or 1 hour run time for identification of single species or complex sample, respectively. Mass spectrometry data were processed in Scaffold v4.8.3 (Proteome Software, Inc.). A protein threshold of 99% minimum with two peptides minimum and peptide threshold of 1% false discovery rate was applied. The data were exported, and custom Python 2.7 scripts were used to calculate the relative spectral abundance factor (RSAF) for each protein by dividing the total number of spectral counts by the molecular weight. Values for each protein were normalized to the bait, Enp1, to reflect relative stoichiometry. S2 Table contains relevant spectral counts and processed data from the mass spectrometry experiments.

## Sucrose density gradient analysis

For polysome profile analysis of the suppressors of *bud23Δ*, BY4741, AJY2676, AJY3744, AJY4529, AJY4531, and AJY4535 were cultured overnight in YPD to saturation. Cultures were diluted into YPD at a starting OD$_{600}$ of 0.02 and cultured to early exponential phase (OD$_{600}$~0.10–0.13) and then treated with cycloheximide (CHX) at 100 µg/ml for 10 minutes at 30°C to inhibit translation. After centrifugation cells were frozen in liquid nitrogen and stored at -80°C. Cells were washed and resuspended in Lysis Buffer (50 mM Tris-HCl pH 7.6 (25°C), 100 mM KCl, 5 mM MgCl$_2$, 7 mM βME, 100 µg/mL CHX, 1 mM each of PMSF and Benzamidine, and 1 µM each of leupeptin and pepstatin). Extracts were prepared by glass bead lysis and clarified by centrifugation for 15 minutes at 18,000*g* at 4°C. 4.5 A$_{260}$ units of clarified extract were loaded onto 7–47% sucrose gradients made in the same buffer lacking protease inhibitors. Gradients were subjected to ultracentrifugation for 2.5 hours at 40,000 rpm in a Beckman SW40 rotor. The gradients were subjected to continuous monitoring at 254 nm using an ISCO Model 640 fractionator.

For analysis of the sedimentation of factors in the absence of Bud23, AJY2665 and AJY4395 were cultured overnight to saturation. Cells were diluted into YPD at a starting OD$_{600}$ of 0.05 and cultured for three hours (OD$_{600}$ = ~0.08–0.1). Cultures were treated with 0.5 mM auxin for 2 hours at 30°C, then treated with CHX at 100 µg/mL for 10 minutes at 30°C. Cells were harvested and stored as described above. Cells were washed and resuspended in Lysis Buffer supplemented with an EDTA-free Pierce Protease Inhibitor Mini Tablet cocktail (Thermo Scientific). Extracts were generated, and nine A$_{260}$ units were loaded onto sucrose gradients and subject to ultracentrifugation as described above. Gradients were fractionated into 600 µL

fractions with continuous monitoring at 254 nm using an ISCO Model 640 fractionator. Proteins were precipitated using 15% TCA as described previously [33]. Proteins from 10% of each fraction were separated on SDS-PAGE gels and subjected to Western blotting.

## Western blot analysis

Primary antibodies used in this study were anti-c-Myc monoclonal 9e10 (Biolegend), anti-HA (Biolegend), anti-Bud23 (C. Wang), anti-Rps24 (our laboratory), anti-Glucose-6-phosphate dehydrogenase (Sigma ImmunoChemicals), and anti-Imp4 (S. Baserga). Secondary anti-bodies were goat anti-mouse antibody-IRDye 800CW (Li-Cor Biosciences), goat anti-rabbit anti-body-IRDye 680RD (Li-Cor Biosciences), and goat anti-rabbit antibody-HRP (Jackson Immunoresearch Laboratories). The blots in Figs 3D, 3F, 5C, and 6B were imaged with an Odyssey CLx infrared imaging system (Li-Cor Biosciences) using Image Studio (Li-Cor Biosciences). The blots in Fig 6C and S4C Fig were imaged using SuperSignal West Pico PLUS Chemiluminescent Substrate (Thermo Scientific) and exposed to film.

## Molecular visualization

All images of SSU Processome and Dhr1 structures are from PDB ascension codes 5WLC and 6H57, respectively, unless otherwise noted. The images of the pre-40S and Bud23-Trm112 subcomplex structures are from PDB accession code 6G4W. The structures of GDPNP- and GDP-bound EF-Tu are from PDB ascension codes 1B23 and 1EFC, respectively. Molecular visualizations were generated using MacPyMOL: PyMOL v1.8.2.1 Enhanced for Mac OS X (Schrödinger LLC) unless indicated otherwise.

## Supporting information

**S1 Fig. Schematic of rRNA processing relevant to 40S production in *S. cerevisiae*.** Endonu-cleolytic processing of the pre-18S rRNA at sites A0, A1, and A2 (or A3) occurs within the context of the SSU Processome. Events occurring co- or post-transcriptionally are denoted by blue and red scissors, respectively. Cleavage at either the A2 or A3 sites liberates the SSU precursors from the LSU precursors containing the 27SA2 and 27SA3 rRNA intermediates, respectively. In rapidly dividing cells processing of A0, A1, and A2 appear to occur co-transcriptionally in a sequential order to produce the 20S rRNA intermediate. The RNA Exosome exonucleolytically degrades the A0- and A1-cleaved 5' ETS fragments. When A1 or A2 cleavage is delayed, post-transcriptional cleavage at A3 occurs to produce the 22S or 21S rRNA, and A1 and A2 cleavage instead occur post-transcriptionally to generate the 20S rRNA. When SSU Processome function is entirely precluded, cleavage at A3 produces the 23S rRNA that becomes degraded. The 20S rRNA is a component of pre-40S intermediates that are exported to the cytoplasm where an endonuclease processes the D site to yield the 18S rRNA.
(TIFF)

**S2 Fig. Secondary structure diagram of the 18S rRNA.** The 18S rRNA is divided into four main domains: the 5' domain (blue), central domain (gold), 3' major domain (purple and black), and the 3' minor domain (green). The 3' basal subdomain (black) is a sub-region of the 3' major domain that forms during the assembly of the SSU Processome [17], and contains the binding site for Bud23. The base methylated by Bud23, guanosine 1575 (G1575, red) is indicated. The position of the central pseudoknot (CPK, gray) is also pictured.
(EPS)

**S3 Fig. The position of Utp14 and the binding site of Dhr1 within the SSU Processome.** (A) The location of the resolved segments of Utp14 (brown) in the SSU Processome. A contour

line indicates the unresolved region of Utp14 where the Dhr1-interaction surface and *bud23Δ*-suppressing mutations are located. The U3 snoRNA binding site of Dhr1 and U3 mutations that suppress a cold-sensitive Dhr1 mutant [29] are indicated by cyan and black sticks, respectively. Bms1, Imp4, Rps28, Utp2, and the 3' basal subdomain RNA are shown for reference. (B) A cartoon of Utp14 primary structure indicating the position of its resolved portions and the *bud23Δ*-suppressing mutations reported here (black) and previously (light blue) within its Dhr1-activation loop [30].
(TIFF)

**S4 Fig. Analysis of combinatorial Imp4 mutants.** (A) Complementation by the indicated *IMP4* alleles as shown by 10-fold serial dilutions of wild-type cells (BY4741) or $P_{GAL1}$-*3xHA-IMP4* (AJY3822) cells harboring either an empty vector (pRS315) or vectors encoding the indicated alleles of *IMP4* spotted on SD-Leu media containing galactose or glucose and grown for two days at 30˚C. (B) Suppression of the growth defect of *bud23Δ* by the ectopic expression of the indicated *IMP4* alleles as shown by 10-fold serial dilutions of wild-type cells (BY4741) or *bud23Δ* (AJY2676) cells harboring either an empty vector (pRS315) or vectors encoding the indicated alleles of *IMP4* spotted on SD-Leu- media containing glucose and grown for two days at 30˚C. (C) Left panel: Complementation of Imp4-13xmyc as shown by 10-fold serial dilutions of BY4741 cells or $P_{GAL1}$-*3xHA-IMP4* (AJY3822) cells harboring either an empty vector (pRS315) or vectors expressing tagged or untagged versions of the indicated *IMP4* alleles spotted on SD-Leu- media containing glucose and grown for two days at 30˚C. Right panel: The sucrose density gradient sedimentation of ectopically expressed Imp4-13xMYC and Imp4-TSMD-13xMYC. Extracts were prepared from BY4741 cells harboring vectors expressing Imp4-13xMYC (pAJ2720) or Imp4-TSMD-13xMYC (pAJ2723) as described in the Materials and methods of the main text except 150 μg/ml CHX was used. For each sample, 9 $A_{260}$ units of clarified extract was separated on sucrose density gradients prior to fractionation. Proteins from each fraction were precipitated with TCA and subjected to Western blot analysis using anti-c-myc antibody (Covance).
(TIFF)

**S5 Fig. Complementation of *BMS1* alleles in *BUD23*-replete cells.** Complementation of selected *BMS1* alleles in *BUD23*-replete cells as shown by 10-fold serial dilutions of BY4741 cells and $P_{TETOFF}$-*BMS1* (AJY4377) cells harboring either an empty vector (pRS415) or vectors encoding the indicated *BMS1* alleles spotted on SD-Leu- media containing glucose and 20 μg/mL doxycycline and grown for three days at 30˚C.
(TIFF)

**S6 Fig. Comparison of the structure of Bms1 to the conformational states of EF-Tu.** (A) Structural alignment of EF-Tu bound to the non-hydrolysable GTP analog, GDPNP (slate blue, PDB 1B23) to domain I of Bms1 (from PDB 5WLC) is shown. Bms1 is colored by domains as in Fig 4D; the GTP analog and magnesium ion bound to EF-Tu are shown as orange sticks and green sphere. Structures are shown individually (left, middle) and as an overlay (right). (B) A view of domains II and III of Bms1 compared to those of EF-Tu shows that the two domains adopt beta barrels in similar conformations. (C) GDPNP-bound EF-Tu forms a complex with tRNA, while GDP-bound EF-Tu (deep purple, PDB 1EFC) does not. (D) Conformational differences in the beta-barrel domains of GDP and GTP-bound EF-Tu suggest that these domains rotate away from one another upon GTP hydrolysis to promote tRNA release. (E) The amino-acyl tRNA contacts GDPNP-bound EF-Tu through its two beta barrel domains. (F) Bms1 in the same orientation as EF-Tu in panel E. The unstructured loop of domain V that connects it to domain IV (denoted by the black arrow) and an N-terminal

helix of Mpp10 (red) contacts domains II and III of Bms1 in a manner reminiscent of how tRNA interacts with GDPNP-bound EF-Tu. The mutated residues that suppress *bud23Δ* are shown as magenta sticks.
(TIFF)

**S7 Fig. Complementation of *DHR1* alleles in *BUD23*-replete cells.** Complementation of select *DHR1* alleles in *BUD23*-replete cells as shown by 10-fold serial dilutions of $P_{GAL1}$-*3xHA-DHR1*, $P_{GAL1}$-*3xHA-UTP14* (AJY4605) cells harboring a vector expressing *UTP14* (pAJ1919; [45]) and either an empty vector (pRS415) or vectors encoding the indicated *DHR1* alleles spotted on SD-Leu-Ura- media containing glucose and grown for two days at 30˚C.
(TIFF)

**S8 Fig. Proteomic compositions of 40S pre-cursors from cells with or without Bud23.** Related to Fig 7. A heatmap of SSU biogenesis proteins that co-immunoprecipitated with Enp1-TAP in the presence (+) or absence (-) of Bud23 is shown. The scale spanning from 0 (white) to 1 (cyan) reflects the relative spectral abundance factor (RSAF). The RSAF was calculated by first normalizing the total number of spectral counts identified for a given protein to its molecular weight; these values were further normalized to the bait, Enp1, to reflect stoichiometry. RSAF values for each protein are shown within each cell. For each protein, the number of spectral counts identified in the presence or absence of Bud23 are shown in parentheses, respectively. Proteins that showed a significant increase or decrease relative to the + Bud23 sample and are listed in S9 Fig are denoted by an asterisks (*) colored blue or black, respectively. Proteins are grouped according to [7] or by known function. Heatmaps were generated in Graphpad Prism version 8.3.0 (328) for Mac iOS. The complete data for this figure are available in S2 Table.
(TIFF)

**S9 Fig. 40S biogenesis factors whose association with 40S pre-cursors significantly changed upon Bud23-depletion.** Related to Fig 7 and S8 Fig. Mass spectrometry analysis of total proteins that co-precipitated with Enp1. Proteins that showed a significant log2 fold-change difference in the absence or presence or Bud23 are shown. Total number of peptides identified for each protein was normalized to molecular weight then further normalized to the bait to generate RSAF values (see Materials and methods) which were used to calculate the log2 fold-change between the mutant and wild-type samples. Proteins displaying a ± 0.5-fold change or more with a difference of greater than 10 total spectral counts are plotted. Proteins are grouped according to when they first bind to pre-ribosomes [7]. The complete mass spectrometry data are available in S2 Table.
(TIFF)

**S10 Fig. Dhr1 and Bms1 physically interact in the Dis-C complex.** The GTPase core of Bms1 (green) and the helicase core of Dhr1 (dark gray) are opposite one another in the Dis-C complex (PDB 6ZQG), but part of the N-terminal domain (NTD) of Dhr1 interacts with Bms1 on the distal side of the U3-18S heteroduplexes (magenta/gold) that Dhr1 unwinds. The 3' basal subdomain (light gray), Imp4 (blue), Utp2 C-terminal domain (CTD; orange), and Utp14 (brown) are shown for reference.
(TIFF)

**S1 Table. Cumulative list of the suppressors of *bud23Δ*.**
(DOCX)

**S2 Table. Mass spectrometry data for the Enp1-TAP affinity purifications.**
(XLSX)

## Acknowledgments

We thank S. Baserga and C. Wang for the Imp4 and Bud23 antibodies, respectively. The vector pJW1662 was a gift from J. Weissman (Addgene plasmid #112050). We thank J. Ream and J. Zhu for assistance with cloning, and P. Sujita for assistance with preparing samples for Illumina sequencing submission. We also thank the members of the A. Johnson and Sarinay-Cenik laboratories for helpful comments regarding the manuscript.

## Author Contributions

**Conceptualization:** Joshua J. Black, Richa Sardana, Arlen W. Johnson.

**Formal analysis:** Joshua J. Black.

**Funding acquisition:** Arlen W. Johnson.

**Investigation:** Joshua J. Black, Richa Sardana, Ezzeddine W. Elmir.

**Project administration:** Arlen W. Johnson.

**Supervision:** Joshua J. Black, Arlen W. Johnson.

**Visualization:** Joshua J. Black.

**Writing – original draft:** Joshua J. Black, Arlen W. Johnson.

**Writing – review & editing:** Joshua J. Black, Arlen W. Johnson.

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
