## [Decision Letter · Decision Letter 0]

29 Jul 2020

Dear Arlen,

Thank you very much for submitting your Research Article entitled 'Bud23 promotes the progression of the Small Subunit Processome to the pre-40S ribosome in Saccharomyces cerevisiae' to PLOS Genetics. Your manuscript was fully evaluated at the editorial level and by three independent peer reviewers. The reviewers appreciated the attention to an important topic but identified some aspects of the manuscript that should be improved.

We therefore ask you to modify the manuscript according to the review recommendations before we can consider your manuscript for acceptance. Your revisions should address the specific points made by each reviewer, as summarized in my comments to you.

"The main thrust of your argument is that each of the mutations destabilizes the 90S intermediate. If so, one would expect that these mutations might have growth defects by themselves. For a subset of mutants this is shown. I am certain that you have these data, and think it would be useful to include them.

To more directly assess whether individual proteins are destabilized, it would be terrific if you could show that one or two of the mutant protein sediment outside of pre-ribosomes. as a proof that this is true...i dont think it would be necessary to do this with each mutant, or each protein....one or two proteins (whatever you have a good antibody for) in one or two mutants...(with wt bud23)."

[LINK]

Yours sincerely,

Katrin P. Karbstein, Ph.D.

Guest Editor

PLOS Genetics

Gregory Copenhaver

Editor-in-Chief

PLOS Genetics

Dear Arlen,

your manuscript has now been evaluated by three reviewers as well as by myself. I am assuming that the comments will be appended...

Two of the reviewers are very favorable, and they have seen previous iterations.

A third reviewer has a couple of extra comments, which I think are worth considering.

The main thrust of your argument is that each of the mutations destabilizes the 90S intermediate. If so, one would expect that these mutations might have growth defects by themselves. For a subset of mutants this is shown. I am certain that you have these data, and think it would be useful to include them.

To more directly assess whether individual proteins are destabilized, it would be terrific if you could show that one or two of the mutant protein sediment outside of pre-ribosomes. as a proof that this is true...i dont think it would be necessary to do this with each mutant, or each protein....one or two proteins (whatever you have a good antibody for) in one or two mutants...(with wt bud23)...

Stay safe!

Best,

Katrin

Reviewer's Responses to Questions

**Comments to the Authors:**

Reviewer #1: Here Black et al. have devised a series of genetic and biochemical studies to elucidate the role of Bud23 in the disassembly of the small subunit processome. By screening for suppressors of a Bud23 deletion, the authors have identified a number of suppressor mutations in a set of assembly factors that functionally connect the 3’ domain with the unwinding of the U3-18S RNA duplexes. The authors propose a model in which the binding of Bud23 to pre-rRNA aids in the disassembly of additional ribosome assembly factors from the SSU processome. This model also postulates that in the absence of Bud23, mutations that weaken protein-protein interactions or reduce productive enzymatic activities (GTPase or RNA helicase activities by Bms1 and Dhr1 respectively) can compensate for an otherwise suboptimal compaction of the SSU processome into a pre-40S particle. This in-depth study provides important inroads for subsequent structural and mechanistic studies of the SSU processome and is of general interest for researchers studying RNA-protein interactions and larger assemblies.

It is the opinion of this reviewer, that the authors have addressed all comments that were raised during a previous round of review and that provided that the minor comment listed below is addressed, this manuscript should be published in PLOS Genetics.

Minor comment:

In figure 6C the authors show that in response to the depletion of Bud23, the assembly factor Enp1 is largely shifted to lower molecular weight species while Imp4 is still present at a reduced level in higher molecular weight complexes. This would suggest that there are different sets of complexes resolved on the sucrose gradient that are currently not identified. In support of this, the purified RNPs containing Enp1 (Fig. 7A) in the presence or absence of Bud23 further indicate that several additional ribosome assembly factors (including Kre33, Utp2 and Bfr2 among others) are enriched upon depletion of Bud23. Figure 6C therefore only provides an incomplete picture of the composition and complexity of the mixture of particles that are contained within the Bud23 depleted sample shown in Fig.7A.

To address this, the authors should provide Western blots for enriched proteins (Kre33, Utp2, Bfr2) or reduced proteins (Tsr1 or Rio2) for figure 6c.

Reviewer #2: I have previously reviewed this paper for another journal and my specific comments have been suitably addressed.

The present paper continues a series of genetic analyses based on the observation that bud23∆ mutants generate spontaneous suppressors in other early-acting ribosome synthesis factors. The authors isolate and identify many new suppressor mutations in known targets, as well as in two genes not previously picked up in this screen. They then make good use of the published pre-ribosome structure data to interpret the mutations and design specific mutations to test specific, predicted interactions. This allows them to make some interesting and plausible predictions on the role of Bud23 in helping dissociate factors from the 3' minor domain at the SSU processome => pre-40S transition. These are good genetic analyses although, as the authors point out, biochemistry and structural analyses will be needed to validate the hypotheses.

Figs. 7, S5, S6 - provide clearest data in MS for requirement for Bud23 during SSU processome disassembly. It is less clear how direct/active is this function, but the data support the existence of an intricate network of interactions needed for correct assembly and timely disassembly of the processome. These are nicely presented in Fig. 8.

The experimental work is technically solid and the conclusions offer several significant starting hypotheses for future work. Overall, the work appears well suited to publication in PLoS Gen.

Reviewer #3: In this manuscript the Johnson laboratory nicely takes benefit of recent structural achievements in the ribosome biogenesis field to better rationalize the structural/functional basis of extra-genic suppressors of bud23 deletion, a non-essential trans-acting factor involved in SSU biogenesis.

Based on this and additional analyses, the authors proposed a model where Bud23 promotes disruption of rRNA/protein and protein/protein interactions network, thereby leading to the release of several ribosome biogenesis contributing to this network and enabling SSU progression/transition to later steps of SSU assembly.

Whereas I find the observation very interesting, I am concerned about the lack of sufficient biochemical analysis convincingly supporting the major implication of the authors interpretation. I feel that the authors should provide more compelling mechanistic insights going beyond their valid, but not validated, hypothesis.

Specific comments:

1) The main conclusion of this study, is that the bud23 extra-genic suppressor mutations may individually decrease the interaction properties of the protein/protein and protein/rRNA interaction network, enabling the “collective” release of ribosome biogenesis factors contributing to this interaction network.

One implication is that independently of the nature of the mutation and affected factors, each individual suppressing mutation event is sufficient to modify the biochemical property of the full interaction network as such that it can bypass Bud23 function. In other words, whereas diverse, the suppressor mutations are functionally equivalent. It is interesting, as it is conceptually very different than a cooperative effect due to suppressor mutations distributed across an interaction network that would sum-up and collectively facilitate progression of SSU maturation in the absence of Bud23.

However, the experimental basis to better understand this intriguing observation and validate the postulated hypothesis is not fully convincing.

Additional efforts should be made to directly substantiate the decrease interactions of exemplary mutated ribosome biogenesis factors and members of the interaction network from purified pre-ribosomal subunits in a bud23 null background. For example, using the implicated factors as bait would clarify their direct and respective behaviour along the maturation pathway. Similarly, what are the consequences of introducing exemplary individual suppressor mutation in a Bud23 wildtype background? Are there any biochemical consequences on the interaction behaviour of this interaction network with pre-ribosomal particles? I would somehow expect a similar biochemical perturbation on the proposed interaction network. What happens to Bud23 in this case, does it get partially “excluded” from the maturation process?

2) Unless, I got this wrong, the experimental design presented in Figure 5 is not very helpful. Considering the bud23 suppressors are individual/unique events (see also above), why testing combination of multiple mutations found in Dhr1 for these interaction analyses? Are the single mutations not sufficient to modify the Dhr1/Utp14 binding property?

**Have all data underlying the figures and results presented in the manuscript been provided?**

Reviewer #1: Yes

Reviewer #2: Yes

Reviewer #3: Yes

PLOS authors have the option to publish the peer review history of their article (what does this mean?). If published, this will include your full peer review and any attached files.

Reviewer #1: **Yes: **Sebastian Klinge

Reviewer #2: **Yes: **David Tollervey

Reviewer #3: No

---

## [Decision Letter · Decision Letter 1]

27 Sep 2020

Dear Arlen,

Thank you very much for submitting your Research Article entitled 'Bud23 promotes the progression of the Small Subunit Processome to the pre-40S ribosome in Saccharomyces cerevisiae' to PLOS Genetics.

As outlined in my "review" I believe it would benefit this work, its impact and "longevity" greatly if you considered it in light of the new structures.

[LINK]

Yours sincerely,

Katrin Karbstein, Ph.D.

Guest Editor

PLOS Genetics

Gregory Copenhaver

Editor-in-Chief

PLOS Genetics

Dear Arlen,

thank you for resubmitting your manuscript.

We have now received the comments from the previous reviewer that had voiced concerns.

As you can see there are still some concerns. Nonetheless, given the new structures from the Beckmann and Ye labs I feel it is important to get your work out.

Nonetheless, I also suspect that the impact of the work would benefit from adding the information from these structures into the manuscript (and potentially revising details of your model). I want to very strongly urge you to do this because i think it will greatly increase the longevity of this study, and its impact. Naturally, this does not require any more experimentation, and I suspect you have already done this now that the pdb coordinates are released.

Best,

Katrin

Reviewer's Responses to Questions

**Comments to the Authors:**

Reviewer #3: I sincerely appreciate the efforts made by the authors to provide additional results and to take the comments/suggestions into consideration, however, the functional and biochemical analysis, as analysed so far, do not provide the necessary compelling and logical evidences supporting the interpretation and model proposed by the authors of how Bud23 may fulfil its function.

There is no doubt that the data are of high standard and in case I have misunderstood the authors’ interpretations and their biological implications, I hope they will make sufficient efforts to alleviate the possible source(s) of this misunderstanding, if any.

As indicated in my previous comments, there are significant logical gap and inconsistencies which still needs to be fully addressed or the proposed model needs to be substantially revised to reflect the available data. Based on the current experimental evidence the proposed model is not very intuitive and the arguments are not fully convincing (at least to me).

In the case that the authors have not fully understood my previous comments/suggestions and to alleviate any ambiguity, I have reformulated my main concerns below in order to complement and clarify, if necessary, my previous comments.

The main logical problem in the authors argumentation is to reconcile the fact that individual point mutations in different assembly factor/r-protein bypass the bud23 growth-defect and presumably its function independently of each other’s. In fact, based on the nature and structural environment of the observed extra-genic suppressors, the authors interpret their genetic interactions study in a more general physical-interaction network and accordingly conclude that Bud23 influence the disassembly/stability of this physical (genetic) network. The implication of this interpretation is that the suppressor mutations individually contribute to biochemical perturbation of this physical interaction network.

Considering that the genetic suppression is based on single and individual mutation events across various factors. This implies that any of these should have some common/similar functional effects/consequences. According to the authors: destabilization of protein/protein interaction protein/RNA interactions. Among the examplary anaylysis that would/could support this interpretation, the combination of multiple “suppressor mutations” are necessary to observe the expected functional consequences in agreement with the author’s model (e.g. Dhr1-Utp14 interactions). However, and in contracdiction to the proposed model, when isolated, the individual mutations have no apparent biochemical consequences (e.g. Dhr1-Utp14 interactions/rebuttal).

Strikingly, combining suppressor mutations, when tested, are not able to suppress the Bud23 loss of function anymore and do not show obvious differences in their biochemical behaviour, when compare to WT (e.g. Imp4 analysis Supp Fig4).

An additional (new) argumentation line or alternative (?) explanation is that the extensive interaction network within the 90S would somehow sustain the interaction to “normal” level or the changes might be too subtle to be measured using “standard conditions”. This could be true, but this remains a very vague justification and rather contradicting with the advertised authors interpretation. Challenging the interactions strength of exemplary individual suppressor mutations and/or the bud23-network members to pre-ribosome using more stringent biochemical conditions (e.g. increasing salt concentrations) could help to validate this possibility.

**Have all data underlying the figures and results presented in the manuscript been provided?**

Reviewer #3: Yes

PLOS authors have the option to publish the peer review history of their article (what does this mean?). If published, this will include your full peer review and any attached files.

Reviewer #3: No

---

## [Editor Report · Decision Letter 2]

21 Oct 2020

Dear Arlen,

Thank you so much for your careful revision of this manuscript. We are pleased to inform you that your manuscript entitled "Bud23 promotes the final disassembly of the Small Subunit Processome in Saccharomyces cerevisiae" has been editorially accepted for publication in PLOS Genetics. I am so excited to see this paper out finally, and really believe that these new structures make it even more relevant! Congratulations!

Yours sincerely,

Katrin Karbstein, Ph.D.

Guest Editor

PLOS Genetics

Gregory P. Copenhaver

Editor-in-Chief

PLOS Genetics

Comments from the reviewers (if applicable):

**Data Deposition**

http://datadryad.org/submit?journalID=pgenetics&manu=PGENETICS-D-20-00852R2

**Press Queries**

---

## [Editor Report · Acceptance letter]

19 Nov 2020

PGENETICS-D-20-00852R2 

Bud23 promotes the final disassembly of the Small Subunit Processome in Saccharomyces cerevisiae 

Dear Dr Johnson, 

We are pleased to inform you that your manuscript entitled "Bud23 promotes the final disassembly of the Small Subunit Processome in Saccharomyces cerevisiae" has been formally accepted for publication in PLOS Genetics! Your manuscript is now with our production department and you will be notified of the publication date in due course.

With kind regards,

Nicola Davies

PLOS Genetics

On behalf of:
